# Towards an Information Theoretic Framework of Context-Based Offline Meta-Reinforcement Learning

**Lanqing Li**[1,2*], **Hai Zhang**[3*], **Xinyu Zhang**[4], **Shatong Zhu**[3], **Yang Yu**[2],
**Junqiao Zhao**[3†], **Pheng-Ann Heng**[2]
[1] Zhejiang Lab, [2] The Chinese University of Hong Kong,
[3] Tongji University, [4] Stony Brook University

lanqingli1993@gmail.com, {zhanghai12138, zhushatong, zhaojunqiao}@tongji.edu.cn,
zhang146@cs.stonybrook.edu, {yangyu, pheng}@cse.cuhk.edu.hk

## Abstract

As a marriage between offline RL and meta-RL, the advent of offline meta-reinforcement learning (OMRL) has shown great promise in enabling RL agents to multi-task and quickly adapt while acquiring knowledge safely. Among which, context-based OMRL (COMRL) as a popular paradigm, aims to learn a universal policy conditioned on effective task representations. In this work, by examining several key milestones in the field of COMRL, we propose to integrate these seemingly independent methodologies into a unified framework. Most importantly, we show that the pre-existing COMRL algorithms are essentially optimizing the same mutual information objective between the task variable $M$ and its latent representation $Z$ by implementing various approximate bounds. Such theoretical insight offers ample design freedom for novel algorithms. As demonstrations, we propose a supervised and a self-supervised implementation of $I(Z; M)$, and empirically show that the corresponding optimization algorithms exhibit remarkable generalization across a broad spectrum of RL benchmarks, context shift scenarios, data qualities and deep learning architectures. This work lays the information theoretic foundation for COMRL methods, leading to a better understanding of task representation learning in the context of reinforcement learning. Given its generality, we envision our framework as a promising offline pre-training paradigm of foundation models for decision making.[1]

## 1 Introduction

The ability to swiftly learn and generalize to new tasks is a hallmark of human intelligence. In pursuit of this high-level artificial intelligence (AI), the paradigm of meta-reinforcement learning (RL) proposes to train AI agents in a trial-and-error manner by interacting with multiple external environments. In order to quickly adapt to the unknown, the agents need to integrate prior knowledge with minimal experience (namely the context) collected from the new tasks or environments, without over-fitting to the new data. This meta-RL mechanism has been adopted in many applications such as games [1, 2], robotics [3, 4] and drug discovery [5].

However, for data collection, classical meta-RL usually requires enormous online explorations of the environments [6, 7], which is impractical in many safety-critical scenarios such as healthcare [8], autonomous driving [9] and robotic manipulation [10]. As a remedy, offline RL [11] enables agents

---

*Equal contribution. This work was primarily done when Hai Zhang worked as an intern at Zhejiang Lab.
†Corresponding Author
[1]Source code: https://github.com/betray12138/UNICORN.git.

to learn from logged experience only, thereby circumventing risky or costly online interactions. Recently, offline meta-RL (OMRL) [12–14] has emerged as a novel paradigm to significantly extend the applicability of RL by "killing two birds in one stone": it builds powerful agents that can quickly learn and adapt by meta-learning, while leveraging offline RL mechanism to ensure a secure and efficient optimization procedure. In the context of classical supervised or self-supervised learning, which is de facto offline, OMRL is reminiscent of the multi-task learning [15], meta-training [6] and fine-tuning [16–18] of pre-trained large models. We envision it as a cornerstone of RL foundation models [19–21] in the future.

Along the line of OMRL research, context-based offline meta-reinforcement learning (COMRL) is a popular paradigm that seeks optimal meta-policy conditioning on the context of Markov Decision Processes (MDPs). Intuitively, the crux of COMRL lies in learning effective task representations, hence enabling the agent to react optimally and adaptively in various contexts. To this end, one of the earliest COMRL algorithms FOCAL [12] proposes to capture the structure of task representations by distance metric learning. From a geometric perspective, it essentially performs clustering by repelling latent embeddings of different tasks while pulling together those from the same task, therefore ensuring consistent and distinguishable task representations.

Despite its effectiveness, FOCAL is reported to be vulnerable to context shifts [22], i.e., when testing on out-of-distribution (OOD) data (Fig. 1). Such problems are particularly challenging for OMRL, since any context shift incurred at test time can not be rectified in the fully offline setting, which may result in severely degraded generalization performance [22, 23]. To alleviate the problem, follow-up works such as CORRO [24] reformulates the task representation learning of COMRL as maximizing the mutual information $I(\boldsymbol{Z}; \boldsymbol{M})$ between the task variable $\boldsymbol{M}$ and its latent code $\boldsymbol{Z}$. It then approximates $I(\boldsymbol{Z}; \boldsymbol{M})$ by an InfoNCE [25] contrastive loss, where the positive and negative pairs are conditioned on the same state-action tuples $(\boldsymbol{s}, \boldsymbol{a})$. Inspired by CORRO, a recently proposed method CSRO [23] introduces an additional mutual information term between $\boldsymbol{Z}$ and $(\boldsymbol{s}, \boldsymbol{a})$. By explicitly minimizing it along with the FOCAL objective, CSRO is demonstrated to achieve the state-of-the-art (SOTA) generalization performance on various MuJoCo [26] benchmarks.

**Contributions** In this paper, following the recent development and storyline of COMRL, we present a Unified Information Theoretic Framework of Context-Based Offline Meta-Reinforcement Learning (UNICORN) encompassing pre-existing methods. We first prove that the objectives of FOCAL, CORRO and CSRO operate as the upper bound, lower bound of $I(\boldsymbol{Z}; \boldsymbol{M})$ and their linear interpolation respectively, which provides a nontrivial theoretical unification of these methods.

Second, by the aforementioned insight and an analysis of the COMRL causal structures, we shed light on how CORRO and CSRO improve context-shift robustness compared to their predecessors by trading off causal and spurious correlations between $\boldsymbol{Z}$ and input data $\boldsymbol{X}$.

Lastly, by examining eight related meta-RL methods (Table 1) concerning their objectives and implementations, we highlight the potential design choices of novel algorithms offered by our framework. As examples, we investigate two instantiated algorithms, one supervised and the other self-supervised, and demonstrate experimentally that they achieve competitive in-distribution and exceptional OOD generalization performance on a wide range of RL domains, OOD settings, data qualities and model architectures. Our framework provides a principled roadmap to novel COMRL algorithms by seeking better approximations/regularizations of $I(\boldsymbol{Z}; \boldsymbol{M})$, as well as new implementations to further combat context shift.

## 2 Method

### 2.1 Preliminaries, Problem Statement and Related Work

We consider MDP modeled as $M = (\mathcal{S}, \mathcal{A}, T, R, \rho_0, \gamma, H)$ with state space $\mathcal{S}$, action space $\mathcal{A}$, transition function $T(\boldsymbol{s}'|\boldsymbol{s}, \boldsymbol{a})$, bounded reward function $R(\boldsymbol{s}, \boldsymbol{a})$, initial state distribution $\rho_0(\boldsymbol{s})$, discount factor $\gamma \in (0, 1)$ and $H$ the horizon. The goal is to find a policy $\pi : \mathcal{S} \rightarrow \mathcal{A}$ to maximize the expected return. Starting from the initial state, for each time step, the agent performs an action sampled from $\pi$, then the environment updates the state with $T$ and returns a reward with $R$. We denote the marginal state distribution at time $t$ as $\mu_\pi^t(\boldsymbol{s})$. The V-function and Q-function are given by

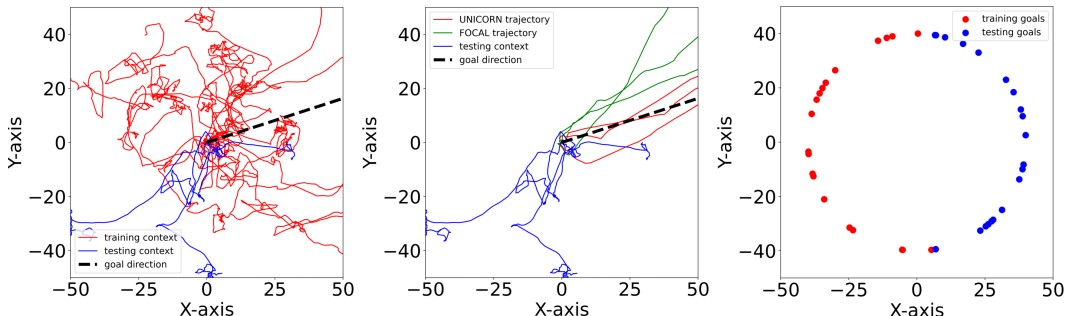

Figure 1: **Context shift of COMRL in Ant-Dir**. **Left**: Given a task $M^i$ specified by a goal direction (dashed line), the RL agent is trained on data generated by a variety of behavior policies trained on the *same* task $M^i$ (red). At test time, however, the context might be collected by behavior policies trained on *different* tasks $\{M^j\}$ (blue), causing a context shift of OOD behavior policies (Section 3.3). **Middle**: Against OOD context, UNICORN (red) is more robust than baselines such as FOCAL (green) in terms of navigating the Ant robot towards the right direction. **Right**: Besides behavior policy, the task distribution (e.g., goal positions in Ant) can induce significant context shift (Section 4.2), which is also a challenging scenario for COMRL models to generalize.

$$V_\pi(\boldsymbol{s}) = \sum_{t=0}^{H-1} \gamma^t \mathbb{E}_{\boldsymbol{s}_t \sim \mu_\pi^t(\boldsymbol{s}), \boldsymbol{a}_t \sim \pi}[R(\boldsymbol{s}_t, \boldsymbol{a}_t)], \tag{1}$$

$$Q_\pi(\boldsymbol{s}, \boldsymbol{a}) = R(\boldsymbol{s}, \boldsymbol{a}) + \gamma \mathbb{E}_{\boldsymbol{s}' \sim T(\boldsymbol{s}'|\boldsymbol{s}, \boldsymbol{a})}[V_\pi(\boldsymbol{s}')]. \tag{2}$$

In this paper, we restrict to scenarios where tasks share the *same* state and action space and can only be distinguished via reward and transition functions. Each OMRL task is defined as an instance of MDP: $M^i = (\mathcal{S}, \mathcal{A}, T^i, R^i, \rho_0, \gamma, H) \in \mathcal{M}$, where $\mathcal{M}$ is the set of all possible MDPs to be considered. For each task index $i$, the offline dataset $X^i = \{(\boldsymbol{s}_j^i, \boldsymbol{a}_j^i, r_j^i, \boldsymbol{s'}_j^i)\}$ is collected by a behavior policy $\pi_\beta^i$. For meta-learning, given a trajectory segment $\boldsymbol{c}_{1:n}^i = \{(\boldsymbol{s}_j^i, \boldsymbol{a}_j^i, r_j^i, \boldsymbol{s'}_j^i)\}_{j=1}^n$ of length $n$ as the context of $M^i$, COMRL employs a context encoder $q_\phi(\boldsymbol{z}|\boldsymbol{c}_{1:n}^i)$ to obtain the latent representation $\boldsymbol{z}^i$ of task $M^i$. If $\boldsymbol{z}$ contains sufficient information about the task identity, COMRL can be treated as a special case of RL on partially-observed MDP [27], where $\boldsymbol{z}$ is interpreted as a faithful representation of the unobserved state. By conditioning on $\boldsymbol{z}$, the learning of universal policy $\pi_\theta(\cdot|\boldsymbol{s}, \boldsymbol{z})$ and value function $V_\pi(\boldsymbol{s}, \boldsymbol{z})$ [28] become regular RL, and optimality can be attained by Bellman updates with theoretical guarantees. To this end, FOCAL [12] proposes to *decouple* COMRL into the upstream *task representation learning* and downstream offline RL. For the former, which can be treated independently, FOCAL employs metric learning to achieve effective clustering of task embeddings:

$$\mathcal{L}_{\text{FOCAL}} = \min_\phi \mathbb{E}_{i,j} \left\{ \mathbb{1}\{i = j\} ||\boldsymbol{z}^i - \boldsymbol{z}^j||_2^2 + \mathbb{1}\{i \neq j\} \frac{\beta}{||\boldsymbol{z}^i - \boldsymbol{z}^j||_2^n + \epsilon} \right\}. \tag{3}$$

However, it is empirically shown that FOCAL struggles to generalize in the presence of context shift [22]. This problem has been formulated in several earliest studies of OMRL. Li et al. [29] observed that when there are large divergence of the state-action distributions among the offline datasets, due to shortcut learning [30], the task encoder may learn to ignore the causal information like rewards and *spuriously correlate* primarily state-action pairs to the task identity, leading to poor generalization performance. Dorfman et al. [14] concurrently identified a related problem which they termed MDP ambiguity in the context of Bayesian offline RL. An example is illustrated in Fig. 1. The problem is further exacerbated in the fully offline setting, as the testing distribution is fixed and cannot be augmented by online exploration, allowing no theoretical guarantee for the context shift.

To mitigate context shift, a subsequent algorithm CORRO [24] proposes to optimize a lower bound of $I(\boldsymbol{Z}; \boldsymbol{M})$ in the form of an InfoNCE [25] contrastive loss

$$\mathcal{L}_{\text{CORRO}} = \min_\phi \mathbb{E}_{\boldsymbol{x}, \boldsymbol{z}} \left[ -\log \left( \frac{h(\boldsymbol{x}, \boldsymbol{z})}{\sum_{M^* \in \mathcal{M}} h(\boldsymbol{x}^*, \boldsymbol{z})} \right) \right], \tag{4}$$

where $\boldsymbol{x} = (\boldsymbol{s}, \boldsymbol{a}, r, \boldsymbol{s}')$, $\boldsymbol{z} \sim q_\phi(\boldsymbol{z}|\boldsymbol{x})$, $h(\boldsymbol{x}, \boldsymbol{z}) = \frac{P(\boldsymbol{z}|\boldsymbol{x})}{P(\boldsymbol{z})} \approx \frac{q_\phi(\boldsymbol{z}|\boldsymbol{x})}{P(\boldsymbol{z})}$, and $\boldsymbol{x}^* = (\boldsymbol{s}, \boldsymbol{a}, r^*, \boldsymbol{s}'^*)$ as a transition tuple generated for task $M^* \in \mathcal{M}$ conditioned on the same $(\boldsymbol{s}, \boldsymbol{a})$. To further combat the *spurious correlation* between the task representation and behavior-policy-induced state-actions, a recently proposed method CSRO [23] introduces a CLUB upper bound [31] of the mutual information between $\boldsymbol{z}$ and $(\boldsymbol{s}, \boldsymbol{a})$, to regularize the FOCAL objective:

$$\mathcal{L}_{\text{CSRO}} = \min_\phi \left\{ \mathcal{L}_{\text{FOCAL}} + \lambda L_{\text{CLUB}} \right\}, \tag{5}$$

$$L_{\text{CLUB}} = \mathbb{E}_i \left[ \log q_\phi(\boldsymbol{z}_i|\boldsymbol{s}_i, \boldsymbol{a}_i) - \mathbb{E}_j \left[ \log q_\phi(\boldsymbol{z}_j|\boldsymbol{s}_i, \boldsymbol{a}_i) \right] \right]. \tag{6}$$

In the next section, we will show how these algorithms are inherently connected and how their context-shift robustness gets improved incrementally.

## 2.2 A Unified Information Theoretic Framework

We start with a formal definition of *task representation learning* in COMRL:

**Definition 2.1** (Task Representation Learning). *Given an input context variable $\boldsymbol{X} \in \mathcal{X}$ and its associated task/MDP random variable $\boldsymbol{M} \in \mathcal{M}$, task representation learning in COMRL aims to find a sufficient statistics $\boldsymbol{Z}$ of $\boldsymbol{X}$ with respect to $\boldsymbol{M}$.*

In pure statistical terms, Definition 2.1 implies that an ideal representation $\boldsymbol{Z}$ is a mapping of $\boldsymbol{X}$ that captures the mutual information $I(\boldsymbol{X}; \boldsymbol{M})$. We therefore define the following dependency structures in terms of directed graphical models:

**Definition 2.2** (Causal Decomposition). *The dependency graphs of COMRL are given by Fig. 2, where $\boldsymbol{X}_b$ and $\boldsymbol{X}_t$ are the behavior-related $(\boldsymbol{s}, \boldsymbol{a})$-component and task-related $(\boldsymbol{s}', r)$-component of the context $\boldsymbol{X}$, with $\boldsymbol{X} = (\boldsymbol{X}_t, \boldsymbol{X}_b)$.*

For the first graph, $\boldsymbol{M} \rightarrow \boldsymbol{X} \rightarrow \boldsymbol{Z}$ forms a Markov chain, which satisfies $I(\boldsymbol{Z}; \boldsymbol{M}|\boldsymbol{X}) = 0$. To interpret the second graph, given an MDP $M \sim \boldsymbol{M}$, the state-action component of $\boldsymbol{X}$ is primarily captured by the behavior policy $\pi_\beta$: $\boldsymbol{s} \sim \mu_{\pi_\beta}(\boldsymbol{s}), \boldsymbol{a} \sim \pi_\beta$. The only exception is when tasks differ in initial state distribution $\rho_0$ or transition dynamics $T$, in which case the state variable $\boldsymbol{S}$ also depends on $\boldsymbol{M}$. We therefore define it as the *behavior-related* component, which should be *weakly* causally related (dashed lines) to $\boldsymbol{M}$ and $\boldsymbol{Z}$. Moreover, when $\boldsymbol{X}_b$ is given, $\boldsymbol{X}_t$ is fully characterized by the transition function $T : (\boldsymbol{s}, \boldsymbol{a}) \rightarrow \boldsymbol{s}'$ and reward function $R : (\boldsymbol{s}, \boldsymbol{a}) \rightarrow r$ of $M$, which should be *strongly*

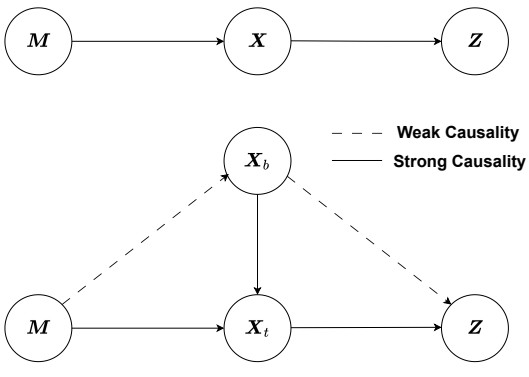

Figure 2: Graphical Models of COMRL.

causally related (solid lines) to $\boldsymbol{M}$ and $\boldsymbol{Z}$ and therefore be defined as the *task-related* component. Mathematically, we find that rewriting $\boldsymbol{X} = (\boldsymbol{X}_t, \boldsymbol{X}_b)$ induces a *causal decomposition* of the mutual information $I(\boldsymbol{Z}; \boldsymbol{X})$ by the chain rule [32]:

$$I(\boldsymbol{Z}; \boldsymbol{X}) = I(\boldsymbol{Z}; \boldsymbol{X}_t|\boldsymbol{X}_b) + I(\boldsymbol{Z}; \boldsymbol{X}_b). \tag{7}$$

We thereby name $I(\boldsymbol{Z}; \boldsymbol{X}_t|\boldsymbol{X}_b)$ and $I(\boldsymbol{Z}; \boldsymbol{X}_b)$ the *primary* and *lesser* causality in our problem respectively. With the setup above, we present the central theorem of this paper:

**Theorem 2.3** (Central Theorem). *Let $\equiv$ denote equality up to a constant, then*

$$\underbrace{I(\boldsymbol{Z}; \boldsymbol{X}_t|\boldsymbol{X}_b)}_{\text{primary causality}} \leq I(\boldsymbol{Z}; \boldsymbol{M}) \leq I(\boldsymbol{Z}; \boldsymbol{X}_t|\boldsymbol{X}_b) + I(\boldsymbol{Z}; \boldsymbol{X}_b) = \underbrace{I(\boldsymbol{Z}; \boldsymbol{X})}_{\text{primary + lesser causality}}$$

*holds up to a constant, where*

1. *$\mathcal{L}_{\text{FOCAL}} \equiv -I(\boldsymbol{Z}; \boldsymbol{X})$.*

2. $\mathcal{L}_{\text{CORRO}} \equiv -I(\boldsymbol{Z}; \boldsymbol{X}_t | \boldsymbol{X}_b)$.

3. $\mathcal{L}_{\text{CSRO}} \geq -\left((1-\lambda)I(\boldsymbol{Z}; \boldsymbol{X}) + \lambda I(\boldsymbol{Z}; \boldsymbol{X}_t | \boldsymbol{X}_b)\right)$.

*Proof.* See Appendix B. □

Theorem 2.3 reveals several key observations. Firstly, the FOCAL and CORRO objectives operate as an upper and a lower bound of $I(\boldsymbol{Z}; \boldsymbol{M})$ respectively. Since one would like to maximize $I(\boldsymbol{Z}; \boldsymbol{M})$ according to Definition 2.1, CORRO, which maximizes the lower bound $I(\boldsymbol{Z}; \boldsymbol{X}_t | \boldsymbol{X}_b)$, can effectively optimize $I(\boldsymbol{Z}; \boldsymbol{M})$ with theoretical assurance. However, FOCAL which maximizes the upper bound $I(\boldsymbol{Z}; \boldsymbol{X}_t, \boldsymbol{X}_b)$ provides *no guarantee* for $I(\boldsymbol{Z}; \boldsymbol{M})$. By Eq. 7, maximizing the FOCAL objective may instead significantly elevate the lesser causality $I(\boldsymbol{Z}; \boldsymbol{X}_b)$, which is undesirable since it contains *spurious correlation* between the task representation $\boldsymbol{Z}$ and behavior policy $\pi_\beta$. This explains *why* FOCAL is less robust to context shift compared to CORRO.

Secondly, CSRO as the latest COMRL algorithm among the three, inherently optimizes a linear combination of the FOCAL and CORRO objectives. In the $0 \leq \lambda \leq 1$ regime, the CSRO objective becomes a convex interpolation of the upper bound $I(\boldsymbol{Z}; \boldsymbol{X})$ and the lower bound $I(\boldsymbol{Z}; \boldsymbol{X}_t | \boldsymbol{X}_b)$ of $I(\boldsymbol{Z}; \boldsymbol{M})$, which in essence, enforces a *trade-off* between the causal ($\boldsymbol{Z}$ with $T$, $\rho_0$) and spurious ($\boldsymbol{Z}$ with $\pi_\beta$) correlation contained in $I(\boldsymbol{Z}; \boldsymbol{X}_b)$. This accounts for the improved performance of CSRO compared to FOCAL and CORRO.

### 2.3 Instantiations of UNICORN

By providing a unified view of pre-existing COMRL algorithms, Theorem 2.3 opens up avenues for novel algorithmic implementations by seeking alternative approximations of the true objective $I(\boldsymbol{Z}; \boldsymbol{M})$. To demonstrate the impact of our proposed UNICORN framework, we discuss two instantiations as follows:

**Supervised UNICORN** $I(\boldsymbol{Z}; \boldsymbol{M})$ can be re-expressed as

$$I(\boldsymbol{Z}; \boldsymbol{M}) = H(\boldsymbol{M}) - H(\boldsymbol{M}|\boldsymbol{Z}) \equiv -H(\boldsymbol{M}|\boldsymbol{Z})$$
$$= \mathbb{E}_{\boldsymbol{z}} \mathbb{E}_{M \sim p(M|\boldsymbol{z})} \left[\log p(M|\boldsymbol{z})\right] = -\mathbb{E}_{\boldsymbol{z}} \left[H(\boldsymbol{M}|\boldsymbol{Z} = \boldsymbol{z})\right]. \tag{8}$$

where $H(\cdot)$ is entropy. Since in practice, each $\boldsymbol{z}^i$ of sample $\boldsymbol{x}^i$ is collected within a specific task $M^i$, minimizing the parameterized entropy $H_{\boldsymbol{\theta}}(\boldsymbol{M}|\boldsymbol{Z} = \boldsymbol{z}^i)$ is equivalent to finding an optimal function $p_{\boldsymbol{\theta}}(M|\boldsymbol{z})$ which correctly assigns the ground-truth label $M^i$ to $\boldsymbol{z}^i$, i.e., optimizing $p_{\boldsymbol{\theta}}(M|\boldsymbol{z})$ towards a delta function $\delta(M - M^i)$ for continuous $\boldsymbol{M}$ or an indicator function $\mathbb{1}(M = M^i)$ for discrete $\boldsymbol{M}$. This implies that

$$\arg\min_{\theta} H_{\boldsymbol{\theta}}(\boldsymbol{M}|\boldsymbol{Z} = \boldsymbol{z}^i) = \arg\max_{\theta} \log p_{\boldsymbol{\theta}}(M^i|\boldsymbol{z}^i). \tag{9}$$

Suppose a total of $n_M$ training tasks $\{M^i\}_{i=1}^{n_M}$ are drawn from the task distribution $p(M)$ with the task label $M$ given for meta-training. Under this supervised scenario, by substituting Eq. 9 into 8, we have

$$\arg\max_{\theta} I(\boldsymbol{Z}; \boldsymbol{M}) = \arg\max_{\theta} \mathbb{E}_{\boldsymbol{z}} \mathbb{E}_M \left[\delta(M - M^i) \log p_{\boldsymbol{\theta}}(M^i|\boldsymbol{z})\right]$$
$$\simeq \arg\max_{\theta} \mathbb{E}_{\boldsymbol{z}} \left[\sum_{i=1}^{n_M} \mathbb{1}(M^i = M) \log p_{\boldsymbol{\theta}}(M^i|\boldsymbol{z})\right], \tag{10}$$

which is precisely the negative cross-entropy loss $H(\boldsymbol{M}, P(\boldsymbol{M}|\boldsymbol{X}))$ for $n_M$-way classification with feature $\boldsymbol{z}$ and classifier $p_{\boldsymbol{\theta}}$. We therefore define the objective of supervised UNICORN as

$$\mathcal{L}_{\text{UNICORN-SUP}} := H(\boldsymbol{M}, P(\boldsymbol{M}|\boldsymbol{X}))$$
$$= -\mathbb{E}_{\boldsymbol{x}, \boldsymbol{z} \sim q_{\boldsymbol{\phi}}(\boldsymbol{z}|\boldsymbol{x})} \left[\sum_{j=1}^{n_M} \mathbb{1}(M^i = M) \log p_{\boldsymbol{\theta}}(M^i|\boldsymbol{z})\right]. \tag{11}$$

Note that $\mathcal{L}_{\text{UNICORN-SUP}}$ is convex and operates as a finite-sample approximation of $-I(\boldsymbol{Z}; \boldsymbol{M})$, for which we derive the following bound:

**Theorem 2.4** (Concentration bound for supervised UNICORN). *Denote by $\hat{I}(\mathbf{Z}; \mathbf{M})$ the empirical estimate of $I(\mathbf{Z}; \mathbf{M})$ by $n_M$ tasks, $\bar{I}(\mathbf{Z}; \mathbf{M})$ the expectation, then with probability at least $1 - \delta$,*

$$\left| \hat{I}(\mathbf{Z}; \mathbf{M}) - \bar{I}(\mathbf{Z}; \mathbf{M}) \right| \leq \sqrt{\frac{\mathrm{Var}(H(\mathbf{Z}|\mathbf{M}))}{n_M \delta}}. \tag{12}$$

*Proof.* See Appendix B. □

Theorem 2.4 bounds the finite-sample estimation error of the empirical risk $\hat{I}(\mathbf{Z}; \mathbf{M})$ with $n_M$ task instances drawn from the real task distribution $p(\mathbf{M})$. The supervised UNICORN has the merit of directly estimating and optimizing the real objective $I(\mathbf{Z}; \mathbf{M})$, which requires explicit knowledge of the task label $M^i$ and a substantial amount of task instances according to Theorem 2.4. For better trade-off of computation and performance, we choose to sample 20 training tasks for all RL environments in our experiments.

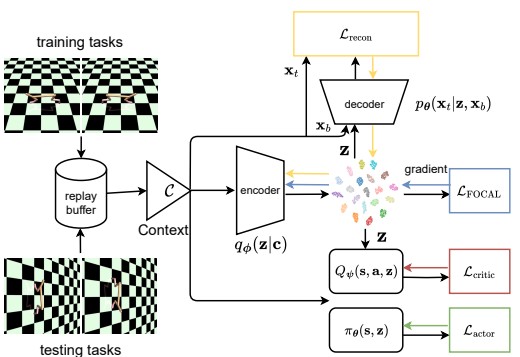

Figure 3: **Meta-learning procedure of UNICORN-SS.** The supervised variant UNICORN-SUP simply replaces the decoder by a classifier $p_{\boldsymbol{\theta}}(M|\mathbf{z})$ and optimize a cross-entropy loss instead of $\mathcal{L}_{\mathrm{recon}}$ and $\mathcal{L}_{\mathrm{FOCAL}}$.

**Self-supervised UNICORN** In practice, offline RL datasets may often be collected with limited knowledge of the task specifications or labels. In this scenario, previous works implement self-supervised learning [33] to obtain effective representation $\mathbf{Z}$, such as the contrastive-based FOCAL/CORRO to optimize $I(\mathbf{Z}; \mathbf{X})/I(\mathbf{Z}; \mathbf{X}_t|\mathbf{X}_b)$ respectively; or generative approaches like VariBAD [34]/BOReL [14] to reconstruct the trajectories $\mathbf{X}$ by variational inference, which is equivalent to maximizing $I(\mathbf{X}_t; \mathbf{Z}, \mathbf{X}_b)$. By Theorem 2.3, these methods optimize a relatively loose upper/lower bound of $I(\mathbf{Z}; \mathbf{M})$, which can be improved by a convex combination of the two bounds:

$$I(\mathbf{Z}; \mathbf{M}) \approx \alpha I(\mathbf{Z}; \mathbf{X}) + (1 - \alpha) I(\mathbf{Z}; \mathbf{X}_t|\mathbf{X}_b), \tag{13}$$

where $0 \leq \alpha \leq 1$ is a hyperparameter. Implementing each term in Eq. 13 allows ample design choices, such as the contrastive losses in Eqs. (3), (4) and (6) or autoregressive generation via Decision Transformer [35, 36] or RNN [34]. For demonstration, in this paper we employ a *contrastive* objective $\mathcal{L}_{\mathrm{FOCAL}}$ as in Eq. (3) for estimating $I(\mathbf{Z}; \mathbf{X})$ while approximate $I(\mathbf{Z}; \mathbf{X}_t|\mathbf{X}_b)$ by reconstruction. By the chain rule:

$$\begin{aligned} I(\mathbf{Z}; \mathbf{X}_t|\mathbf{X}_b) &= I(\mathbf{X}_t; \mathbf{Z}, \mathbf{X}_b) - I(\mathbf{X}_t; \mathbf{X}_b) \\ &\equiv I(\mathbf{X}_t; \mathbf{Z}, \mathbf{X}_b), \end{aligned} \tag{14}$$

since $I(\mathbf{X}_t; \mathbf{X}_b)$ is a constant when $\mathbf{X}_t$ and $\mathbf{X}_b$ are drawn from a fixed distribution as in offline RL. Moreover, by definition of mutual information:

$$\begin{aligned} I(\mathbf{X}_t; \mathbf{Z}, \mathbf{X}_b) &= \mathbb{E}_{\boldsymbol{x_t}, \boldsymbol{x_b}, \boldsymbol{z}} \left[ \log \frac{p(\boldsymbol{x_t}|\boldsymbol{z}, \boldsymbol{x_b})}{p(\boldsymbol{x_t})} \right] \\ &\equiv \mathbb{E}_{\boldsymbol{x_t}, \boldsymbol{x_b}, \boldsymbol{z}} \left[ \log p(\boldsymbol{x_t}|\boldsymbol{z}, \boldsymbol{x_b}) \right] \\ &\geq \mathbb{E}_{\boldsymbol{x_t}, \boldsymbol{x_b}, \boldsymbol{z} \sim q_\phi(\boldsymbol{z}|\boldsymbol{x_t}, \boldsymbol{x_b})} \left[ \log p_{\boldsymbol{\theta}}(\boldsymbol{x_t}|\boldsymbol{z}, \boldsymbol{x_b}) \right], \end{aligned} \tag{15}$$

which induces a *generative* objective $\mathcal{L}_{\mathrm{recon}} := -I(\mathbf{X}_t; \mathbf{Z}, \mathbf{X}_b)$ by reconstructing $\mathbf{X}_t$ with a decoder network $p_{\boldsymbol{\theta}}(\cdot|\boldsymbol{z}, \boldsymbol{x_b})$ conditioning on $\mathbf{Z}$ and $\mathbf{X}_b$. As a result, the proposed unsupervised UNICORN objective can be rescaled as Eq. 16:

$$\mathcal{L}_{\mathrm{UNICORN\text{-}SS}} := \mathcal{L}_{\mathrm{recon}} + \frac{\alpha}{1 - \alpha} \mathcal{L}_{\mathrm{FOCAL}}. \tag{16}$$

The influence of the hyper-parameter $\frac{\alpha}{1-\alpha}$ is shown in Appendix C.4.

Table 1: Comparison between UNICORN instantiations and related existing contextual meta-RL methods. For clarity, "Representation Learning Objective" only lists the loss functions of $Z$ that are independent of the downstream RL tasks. Note that $I(Z; X_t|X_b) \equiv I(X_t; Z, X_b)$ holds *only* for offline RL.

| Method | Setting | Representation Learning Objective | Implementation | Context $X$ |
|---|---|---|---|---|
| **UNICORN-SUP** | Offline | $I(Z; M)$ | Predictive | Transition |
| **UNICORN-SS** | Offline | $\alpha I(Z; X) + (1 - \alpha)I(X_t; Z, X_b)$ | Contrastive+Generative | Transition |
| FOCAL [12, 22] | Offline | $I(Z; X)$ | Contrastive | Transition |
| CORRO [24] | Offline | $I(Z; X_t|X_b)$ | Contrastive | Transition |
| CSRO [23] | Offline | $(1 - \lambda)I(Z; X) + \lambda I(Z; X_t|X_b)$ | Contrastive | Transition |
| GENTLE [41] | Offline | $I(X_t; Z, X_b)$ | Generative | Transition |
| BOReL [14] | Offline | $I(X_t; Z, X_b) - D_{\mathrm{KL}}(q_\phi(Z|X)||p_\theta(Z))$ | Generative | Trajectory |
| VariBAD [34] | Online | $I(X_t; Z, X_b) - D_{\mathrm{KL}}(q_\phi(Z|X)||p_\theta(Z))$ | Generative | Trajectory |
| PEARL [7] | Online | $-D_{\mathrm{KL}}(q_\phi(Z|X)||p_\theta(Z))$ | N/A | Transition |
| ContraBAR [42] | Offline&Online | $I(Z; X_t|A)$ | Contrastive | Trajectory |

We illustrate our learning procedure in Fig. 3 with pseudo-code in Algorithms 1 and 2. A holistic comparison of our proposed algorithms with related contextual meta-RL methods is shown in Table 1. The extra KL divergence in methods like VariBAD and PEARL can be interpreted as the result of a variational approximation to an information bottleneck [37, 38] that constrains the mutual information between $Z$ and $X$, which we found unnecessary in our offline setting (see ablation in Table 6.) Behavior regularized actor critic [39] is employed to tackle the bootstrapping error [40] for downstream offline RL implementation.

## 3 Experiments

For brevity, we name our proposed supervised and self-supervised algorithms UNICORN-SUP and UNICORN-SS respectively. To evaluate their effectiveness, our main experiments are organized to address three primary inquiries regarding the core advantages of UNICORN: (1) How does UNICORN perform on in-distribution tasks? (2) How well can UNICORN generalize to data collected by OOD behavior policies? (3) Can UNICORN outperform consistently across datasets of different qualities? The performance is measured by the average return across 20 testing tasks randomly sampled for each environment.

### 3.1 Experimental Setup

**Data Collection** Following FOCAL, for each task, we train a SAC [43] agent from scratch and collect its replay buffer as the offline dataset. Hence our default data represent a mixed distribution of behavior policies, ranging from random to expert level. Policy checkpoints are also kept for creating various new datasets in Sections 3.3 and 3.4.

We adopt MuJoCo [26] and MetaWorld [3] benchmarks to evaluate our methods, which involve six robotic locomotion environments that require adaptation across reward functions (HalfCheetah-Dir, HalfCheetah-Vel, Ant-Dir, Reach) or across dynamics (Hopper-Param, Walker-Param). We compare UNICORN with six competitive OMRL algorithms, which are categorized as context-based, gradient-based and transformer-based methods:

Table 2: **Average testing returns of UNICORN against baselines on datasets collected by IID and OOD behavior policies.** Results are averaged by 6 random seeds. The best is **bolded** and the second best is underlined.

| Algorithm | HalfCheetah-Dir | | HalfCheetah-Vel | | Ant-Dir | | Hopper-Param | | Walker-Param | | Reach | |
|---|---|---|---|---|---|---|---|---|---|---|---|---|
| | IID | OOD | IID | OOD | IID | OOD | IID | OOD | IID | OOD | IID | OOD |
| UNICORN-SS | **1307±26** | **1296±24** | -22±1 | -94±5 | 267±14 | 236±18 | **316±6** | **304±11** | 419±44 | **407±46** | 2775±241 | 2604±183 |
| UNICORN-SUP | 1296±20 | 1130±76 | -25±3 | **-91±5** | 250±4 | **239±16** | 312±4 | 302±12 | 322±28 | 312±39 | 2681±111 | 2641±140 |
| CSRO | 1180±228 | 458±253 | -28±1 | -102±5 | **276±19** | 233±12 | 310±6 | 301±10 | 310±58 | 279±65 | 2720±235 | **2801±182** |
| CORRO | 704±450 | 245±146 | -37±3 | -112±2 | 148±13 | 120±12 | 283±8 | 272±13 | 277±38 | 213±48 | 2468±175 | 2322±327 |
| FOCAL | 1186±272 | 861±253 | -22±1 | -97±2 | 217±29 | 173±24 | 302±4 | 297±13 | 308±98 | 286±91 | 2424±256 | 2316±303 |
| Supervised | 962±356 | 782±429 | -24±1 | -104±1 | 238±39 | 202±38 | 306±10 | 294±8 | 256±60 | 210±28 | 2489±248 | 2283±205 |
| MACAW | 1155±10 | 450±6 | -56±2 | -188±1 | 26±3 | 0±0 | 218±6 | 205±2 | 141±9 | 130±5 | 2431±157 | 1728±79 |
| Prompt-DT | 1176±40 | -25±9 | -118±66 | -249±21 | 1±0 | 0±0 | 234±5 | 202±5 | 185±9 | 156±17 | 2165±85 | 1896±111 |

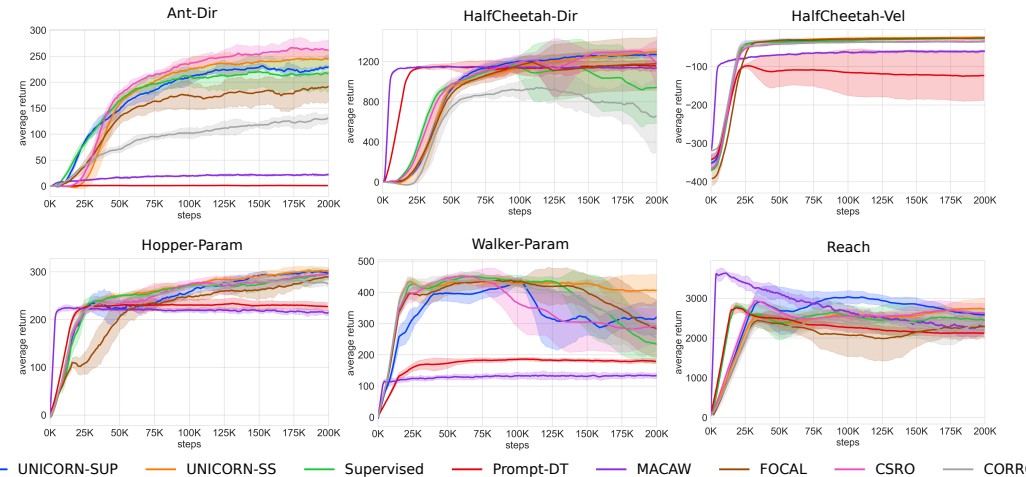

Figure 4: **Testing returns of UNICORN against baselines on six benchmarks.** Solid curves refer to the mean performance of trials over 6 random seeds, and the shaded areas characterize the standard deviation of these trials.

**FOCAL, CORRO, CSRO** are context-based methods that can be seen as special cases or approximations of UNICORN, see details in Section 2.

**Supervised** is a *context-based* method, directly using actor-critic loss to train the policy/value networks and task encoder end-to-end. We find it a competitive baseline across all benchmarks.

**MACAW** [13] is a *gradient-based* method, using supervised advantage-weighted regression for both the inner and outer loops of meta-training.

**Prompt-DT** [44] is a *transformer-based* method, taking context as the prompt of a Decision Transformer (DT) [35, 36] to solve OMRL as a conditional sequence modeling problem.

Our evaluations are performed in a few-shot manner. For gradient-based methods, we first update the meta-policy with a batch of testing context and then evaluate the agent. For transformer-based methods, we utilize trajectory segments as the prompt to condition the auto-regressive rollout.

### 3.2 Few-Shot Generalization to In-Distribution Data

For each environment, the meta-policy is trained on 20 random tasks. To test IID generalization, we sample one trajectory for each testing task as the context, which enables few-shot adaptation of the agent via an updated task representation $z$. Fig. 4 illustrates the learning curves of UNICORN vs. all baselines in terms of testing returns, which correspond to the IID entries of Table 2. The results demonstrate that UNICORN variants, especially UNICORN-SS, consistently achieve SOTA performance across all benchmarks. The observation that the asymptotic performance of CSRO is comparable to UNICORN is expected since it also optimizes a linear combination of the lower and upper bound of $I(\boldsymbol{Z}; \boldsymbol{M})$ by Theorem 2.3, with a different implementation choice. Notably, UNICORN exhibits remarkable stability, especially on HalfCheetah-Dir and Walker-Param, where the other context-based methods suffer a significant performance decline during the later training process. Although the gradient-based MACAW and the transformer-based Prompt-DT show faster convergence, their asymptotic performance leaves much to be desired. Moreover, the average training time for MACAW is about three times longer than the context-based methods.

### 3.3 Few-Shot Generalization to Out-of-Distribution Behavior Policies

Denote by $\{\pi_\beta^{i,t}\}_{i=1:n_M, t=0:n}$ the checkpoints of behavior policies for all tasks logged during data collection, where $i$ refers to the task index and $t$ represents training iteration. To evaluate OOD generalization, for each test task $M^j$, we use *all* behavior policies in $\{\pi_\beta^{i,t}\}_{i=1:n_M, t=0:n}$ to collect the OOD contexts and the conditioned rollouts. The OOD testing performance is measured by averaging returns across all testing tasks and behavior policies, as shown in the OOD entries of Table 2. While all methods suffer notable decline when facing OOD context, UNICORN maintains the strongest

performance by a significant margin. Another observation is that the context-based methods are in general significantly more robust compared to the gradient-based MACAW and transformer-based Prompt-DT. This also justifies the storyline of COMRL development from 2021 to date

$$\text{FOCAL} \rightarrow \text{CORRO} \rightarrow \text{CSRO} \rightarrow \text{UNICORN}$$
$$\text{(2021)} \qquad \text{(2022)} \qquad \text{(2023)} \qquad \text{(2024)}$$

as a roadmap for pursuing more robust and generalizable task representation learning for COMRL.

### 3.4 Influence of Data Quality

To test whether the UNICORN framework can be applied to different data distributions, we collect three types of datasets whose size is equal to the mixed dataset used in Section 3.2 and 3.3: random, medium and expert, which are characterized by the quality of the behavior policies.

As shown in Table 3, UNICORN demonstrates SOTA performance on all types of datasets. Notably, despite CSRO having better IID results than UNICORN on Ant-Dir under the mixed distribution, UNICORN surpasses CSRO by a large margin under the narrow distribution (medium and expert) and random distribution.

We observe that CORRO fails on all three datasets, which is likely attributable to the reliance of CORRO on the negative sample generator. On one hand, under narrow data distribution, there might be little overlap among the high-density state-action regions of datasets from different tasks. On the other hand, samples collected by random policies are often indistinguishable across tasks [22]. Both factors make it more challenging to train a robust generator for producing high quality negative samples conditioning on specific state-action pairs. As for Prompt-DT, its performance improves significantly with increased data quality, which is expected since the decision transformer adopts a behavior-cloning-like supervised learning style.

Table 3: **UNICORN vs. baselines on Ant-Dir datasets of various qualities.** Each result is averaged across 6 random seeds. The best is **bolded** and the second best is underlined.

| Algorithm | Random | | Medium | | Expert | |
|---|---|---|---|---|---|---|
| | IID | OOD | IID | OOD | IID | OOD |
| UNICORN-SS | **81±18** | **62±6** | **220±23** | **243±10** | **279±10** | **262±13** |
| UNICORN-SUP | 75±15 | 60±5 | 140±11 | 126±32 | 247±15 | 229±19 |
| CSRO | 2±3 | 0±1 | 166±10 | 198±17 | 252±39 | 202±45 |
| CORRO | 1±1 | 0±0 | 8±5 | -7±2 | -4±10 | -14±9 |
| FOCAL | 67±26 | 44±10 | 171±84 | 187±86 | 229±42 | 246±20 |
| Supervised | 65±6 | 47±12 | 149±50 | 110±80 | 249±33 | 215±60 |
| MACAW | 3±1 | 0±0 | 28±2 | 1±1 | 88±43 | 1±1 |
| Prompt-DT | 1±0 | 0±0 | 2±4 | 0±1 | 78±15 | 1±2 |

## 4 Discussion

This section presents more empirical evidence on the applicability of the UNICORN framework.

### 4.1 Is UNICORN Model-Agnostic?

As UNICORN tackles task representation learning from a general information theoretic perspective, it is in principle *model-agnostic*. Hence a natural idea is to transfer UNICORN to other model architectures like DT, which we envision to be a promising backbone for large-scale training of RL foundation models.

We employ a straightforward implementation by embedding the task representation vector $z$ as the first token in sequence to prompt the learning and generation of DT, akin to in-context learning for large

Table 4: **DT implementation of COMRL on HalfCheetah-Dir and Hopper-Param.** Each result is averaged by 6 random seeds.

| Algorithm | HalfCheetah-Dir | | Hopper-Param | |
|---|---|---|---|---|
| | IID | OOD | IID | OOD |
| UNICORN-SS | 1307±26 | 1296±24 | 316±6 | 304±11 |
| UNICORN-SS-DT | 1233±10 | 1186±43 | 304±4 | 291±4 |
| UNICORN-SUP-DT | 1227±21 | 1065±57 | 308±6 | 297±2 |
| FOCAL-DT | 1209±33 | 652±36 | 293±4 | 284±5 |
| Prompt-DT | 1177±40 | -25±9 | 234±5 | 203±5 |

language models [45] and in-context RL [46–49]. Instead of the non-trainable prompt used in the original Prompt-DT, we enforce task representation learning of the prompt by optimizing $\mathcal{L}_{\text{UNICORN-SS}}$, $\mathcal{L}_{\text{UNICORN-SUP}}$ and $\mathcal{L}_{\text{FOCAL}}$. We name these variants UNICORN-SS-DT, UNICORN-SUP-DT and FOCAL-DT respectively.

As shown in Table 4 and Fig. 7, applying UNICORN and FOCAL on DT results in significant improvement in both IID and OOD generalization, with UNICORN being the superior option.

Despite the gap between our implementation of UNICORN-DT and its MLP counterpart UNICORN-SS in terms of asymptotic performance, we expect UNICORN-DT to extrapolate favorably in the regime of greater dataset size and model parameters due to the scaling law of transformer [50] and DT [51]. We leave this verification to future work.

## 4.2 Can UNICORN be Exploited for Model-Based Paradigms?

With decoder $p_\theta(x_t|z, x_b)$ as a world model [52], UNICORN-SS can potentially enable a model-based paradigm by generating data using customized latent $z$, which can be especially useful for generalization to OOD tasks. We implement this idea on Ant-Dir by sorting the tasks according to the polar angle of their goal direction ($\theta \in [0, 2\pi)$), and construct task-level OOD by taking 20 consecutive tasks in the middle for training and the rest for testing (Fig. 1). Gaussian noise $\epsilon$ is applied to the real task representation $z$ and $(s, a, z + \epsilon)$ is then fed to the decoder to generate $(s', r)$. The imaginary rollouts

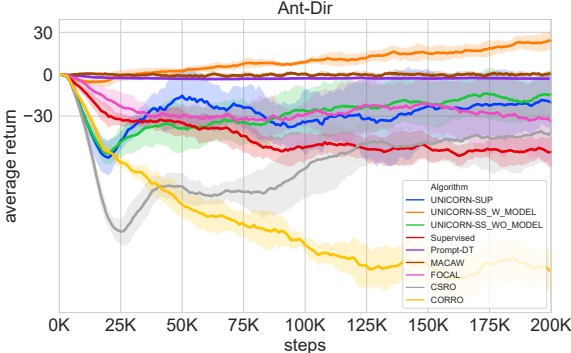

Figure 5: **Testing returns for OOD tasks.** The learning curves are averaged over 6 random seeds.

$\{(s_t, a_t, s'_t, r_t)\}_{t=1}^{n}$ are used during the training of RL agent. We adopt the conventional ensemble technique in model-based RL [53, 54] to stabilize the training process.

As shown in Fig. 5, UNICORN-SS-enabled model-based RL is the only method to achieve positive performance (indicating positive few-shot transfer) in this extremely challenging scenario where context shift is induced by OOD tasks.

## 5 Conclusion & Limitation

In this paper, we unify three signature context-based offline meta-RL algorithms, FOCAL, CORRO and CSRO (potentially a lot more) into a single framework from an information theoretic representation learning perspective. Our theory offers valuable insight on how these methods are inherently connected and incrementally evolved in pursuit of better generalization against context shift. Based on the proposed framework, we instantiate two novel algorithms called UNICORN-SUP and UNICORN-SS which are demonstrated to be remarkably robust and broadly applicable through extensive experiments. We believe our study offers potential for new implementations, optimality bounds and algorithms for both fully-offline and offline-to-online COMRL. Moreover, we expect our framework to incorporate almost all existing COMRL methods that tackle task representation learning. Since according to Definition 2.1, as long as the method tries to solve COMRL by learning a sufficient statistic $Z$ of $X$ w.r.t. $M$, it will eventually come down to optimizing an information-theoretic objective equivalent to $I(Z, M)$, or a lower/upper bound approximation like the ones introduced by Theorem 2.3, up to some regularizations or constraints. With such mathematical generalization as well as the empirical evidence of UNICORN being model-agnostic (see Section 4.1), we envision it as a nascent offline pre-training paradigm of foundation models for decision making [21].

**Limitation** Since our framework assumes a decoupling of task representation learning and offline policy optimization, it does not directly elucidate how high-quality representations are indicative of higher downstream RL performance. Another limitation would be the scale of the experiments. Our offline datasets cover at most 40 tasks and $\sim 450k$ transition tuples, which might be the reason why UNICORN-SUP is inferior to UNICORN-SS ($n_M$ too small to reduce the approximation error in Theorem 2.4). However, we expect our conclusion to extrapolate to larger task sets, dataset sizes and model parameters, for which the DT variants may demonstrate better scaling. Lastly, in principle, the information theoretic formalization in this work should be directly applicable to online RL. However, many of our key derivations rely on the static assumption of $M$ and $X$ (see Appendix B for details), which are apparently violated in the online scenario. Extending our framework to online RL is interesting and nontrivial, which we leave for future work.

## Acknowledgments and Disclosure of Funding

The authors would like to thank the anonymous reviewers for their insightful comments and suggestions. The work described in this paper was supported partially by a grant from the Research Grants Council of the Hong Kong Special Administrative Region, China (Project Reference Number: T45-401/22-N), by a grant from the Hong Kong Innovation and Technology Fund (Project Reference Number: MHP/086/21) and by the National Key Research and Development Program of China (No. 2020YFA0711402).

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

## A Pseudo-Code

---

**Algorithm 1** UNICORN Meta-training

---

**Require:** Offline Datasets $\boldsymbol{X} = \{X^i\}_{i=1}^{N_{\text{train}}}$, training tasks $\boldsymbol{M} = \{M^i\}_{i=1}^{N_{\text{train}}}$, initialized learned policy $\pi_{\boldsymbol{\omega}}$, Q function $Q_{\boldsymbol{\psi}}$, context encoder $q_{\boldsymbol{\phi}}$, classifier/decoder $p_{\boldsymbol{\theta}}$, hyper-parameter $\alpha$, task batch size $N_{tb}$, learning rates $\beta_1, \beta_2, \beta_3, \beta_4$
1: **while** not done **do**
2:     Sample task batch $\{M^j\}_{j=1}^{N_{tb}} \sim \boldsymbol{M}$ and the corresponding replay buffer $\mathcal{R} = \{X^j\}_{j=1}^{N_{tb}} \sim \boldsymbol{X}$
3:     **for** step in each iter **do**
4:         // Train the context encoder and decoder
5:         Sample context $\{\boldsymbol{c}^j\}_{j=1}^{N_{tb}} \sim \mathcal{R}$
6:         Obtain task embeddings $\{\boldsymbol{z}^j \sim q_{\boldsymbol{\phi}}(\boldsymbol{z}|\boldsymbol{c}^j)\}_{j=1}^{N_{tb}}$
7:         Estimate $\mathcal{L}_{\text{UNICORN}}$ ($\mathcal{L}_{\text{UNICORN-SUP}}$ or $\mathcal{L}_{\text{UNICORN-SS}}$)
8:         $\boldsymbol{\phi} \leftarrow \boldsymbol{\phi} - \beta_1 \nabla_{\boldsymbol{\phi}} \mathcal{L}_{\text{UNICORN}}$
9:         $\boldsymbol{\theta} \leftarrow \boldsymbol{\theta} - \beta_2 \nabla_{\boldsymbol{\theta}} \mathcal{L}_{\text{UNICORN}}$
10:         // Train the actor and critic
11:         Detach the task embeddings $\{\boldsymbol{z}^j\}_{j=1}^{N_{tb}}$
12:         Sample training data $\{\boldsymbol{d}^j\}_{j=1}^{N_{tb}} \sim \mathcal{R}$
13:         Estimate $\mathcal{L}_{\text{actor}}, \mathcal{L}_{\text{critic}}$
14:         $\boldsymbol{\omega} \leftarrow \boldsymbol{\omega} - \beta_3 \nabla_{\boldsymbol{\omega}} \mathcal{L}_{\text{actor}}$
15:         $\boldsymbol{\psi} \leftarrow \boldsymbol{\psi} - \beta_4 \nabla_{\boldsymbol{\psi}} \mathcal{L}_{\text{critic}}$
16:     **end for**
17: **end while**

---

**Algorithm 2** UNICORN Meta-testing

---

**Require:** Offline Datasets $\boldsymbol{X} = \{X^i\}_{i=1}^{N_{\text{test}}}$, testing tasks $\boldsymbol{M} = \{M^i\}_{i=1}^{N_{\text{test}}}$, trained policy $\pi_{\boldsymbol{\omega}}$ and context encoder $q_{\boldsymbol{\phi}}$
1: **for** each $M^i$ **do**
2:     Sample context $\boldsymbol{c}^i \sim X^i$
3:     Obtain task embedding $\boldsymbol{z}^i \sim q_{\boldsymbol{\phi}}(\boldsymbol{z}|\boldsymbol{c}^i)$
4:     Rollout policy $\pi_{\boldsymbol{\omega}}(\boldsymbol{a}|\boldsymbol{s}, \boldsymbol{z}^i)$ for evaluation
5: **end for**

---

## B Proofs

### B.1 Proof of Theorem 2.3

We first prove the following lemma:

**Lemma B.1.** *For COMRL, $I(\boldsymbol{Z};\boldsymbol{M}) \geq I(\boldsymbol{Z};\boldsymbol{M}|\boldsymbol{X}_b)$; or equivalently, $I(\boldsymbol{Z};\boldsymbol{M};\boldsymbol{X}_b) \geq 0$, both up to a constant.*

*Proof.*

$$I(\boldsymbol{Z};\boldsymbol{M}|\boldsymbol{X}_t, \boldsymbol{X}_b) = 0$$
$$\Longrightarrow I(\boldsymbol{Z};\boldsymbol{M}) - I(\boldsymbol{Z};\boldsymbol{M}|\boldsymbol{X}_t, \boldsymbol{X}_b) = I(\boldsymbol{M};\boldsymbol{X}_b, \boldsymbol{X}_t) - I(\boldsymbol{M};\boldsymbol{X}_b, \boldsymbol{X}_t|\boldsymbol{Z}) \geq 0$$
$$\Longrightarrow I(\boldsymbol{M};\boldsymbol{X}_b) + I(\boldsymbol{M};\boldsymbol{X}_t|\boldsymbol{X}_b) - I(\boldsymbol{M};\boldsymbol{X}_b|\boldsymbol{Z}) - I(\boldsymbol{M};\boldsymbol{X}_t|\boldsymbol{X}_b, \boldsymbol{Z}) \geq 0$$
$$\Longrightarrow I(\boldsymbol{M};\boldsymbol{X}_b) - I(\boldsymbol{M};\boldsymbol{X}_b|\boldsymbol{Z}) \geq \underbrace{-I(\boldsymbol{M};\boldsymbol{X}_t|\boldsymbol{X}_b)}_{\text{const}} + \underbrace{I(\boldsymbol{M};\boldsymbol{X}_t|\boldsymbol{X}_b, \boldsymbol{Z})}_{\geq 0} \geq \text{const.}$$

Now, since $I(\boldsymbol{Z};\boldsymbol{M}) - I(\boldsymbol{Z};\boldsymbol{M}|\boldsymbol{X}_b) = I(\boldsymbol{M};\boldsymbol{X}_b) - I(\boldsymbol{M};\boldsymbol{X}_b|\boldsymbol{Z})$, we have $I(\boldsymbol{Z};\boldsymbol{M}) - I(\boldsymbol{Z};\boldsymbol{M}|\boldsymbol{X}_b) = I(\boldsymbol{Z};\boldsymbol{M};\boldsymbol{X}_b) \geq \text{const} \equiv 0$. $\qquad\square$

With Lemma B.1, we proceed to prove Theorem 2.3 as follows:

*Proof.* $I(\boldsymbol{Z}; \boldsymbol{X}_t | \boldsymbol{X}_b) \equiv -\mathcal{L}_{\text{CORRO}} \leq I(\boldsymbol{Z}; \boldsymbol{M})$ is shown in the Theorem 4.1. of CORRO [24] and GENTLE [41] with strong assumptions. We hereby present a proof with *no* assumptions. Given $I(\boldsymbol{Z}; \boldsymbol{M}; \boldsymbol{X}_b) \geq 0$ (Lemma B.1), we have

$$
\begin{aligned}
I(\boldsymbol{Z}; \boldsymbol{M}) &= I(\boldsymbol{Z}; \boldsymbol{M} | \boldsymbol{X}_b) + I(\boldsymbol{Z}; \boldsymbol{M}; \boldsymbol{X}_b) \\
&\geq I(\boldsymbol{Z}; \boldsymbol{M} | \boldsymbol{X}_b) \\
&= I(\boldsymbol{M}; \boldsymbol{Z}, \boldsymbol{X}_t | \boldsymbol{X}_b) - I(\boldsymbol{M}; \boldsymbol{X}_t | \boldsymbol{Z}, \boldsymbol{X}_b) \\
&= \underbrace{I(\boldsymbol{Z}; \boldsymbol{M} | \boldsymbol{X}_t, \boldsymbol{X}_b)}_{0 \text{ by } \boldsymbol{M} \to \boldsymbol{X} \to \boldsymbol{Z}} + I(\boldsymbol{M}; \boldsymbol{X}_t | \boldsymbol{X}_b) - I(\boldsymbol{M}; \boldsymbol{X}_t | \boldsymbol{Z}, \boldsymbol{X}_b) \\
&= I(\boldsymbol{M}; \boldsymbol{X}_t | \boldsymbol{X}_b) - I(\boldsymbol{M}; \boldsymbol{X}_t | \boldsymbol{Z}, \boldsymbol{X}_b) \\
&= I(\boldsymbol{M}; \boldsymbol{X}_t | \boldsymbol{X}_b) - H(\boldsymbol{X}_t | \boldsymbol{Z}, \boldsymbol{X}_b) + H(\boldsymbol{X}_t | \boldsymbol{M}, \boldsymbol{Z}, \boldsymbol{X}_b) \\
&\geq I(\boldsymbol{M}; \boldsymbol{X}_t | \boldsymbol{X}_b) - H(\boldsymbol{X}_t | \boldsymbol{Z}, \boldsymbol{X}_b) \\
&= \underbrace{I(\boldsymbol{M}; \boldsymbol{X}_t | \boldsymbol{X}_b)}_{\text{const}} - \underbrace{H(\boldsymbol{X}_t)}_{\text{const}} + I(\boldsymbol{X}_t; \boldsymbol{Z}, \boldsymbol{X}_b) \\
&\equiv I(\boldsymbol{X}_t; \boldsymbol{Z}, \boldsymbol{X}_b) \\
&= I(\boldsymbol{Z}; \boldsymbol{X}_t | \boldsymbol{X}_b) + \underbrace{I(\boldsymbol{X}_t; \boldsymbol{X}_b)}_{\text{const}} \\
&\equiv I(\boldsymbol{Z}; \boldsymbol{X}_t | \boldsymbol{X}_b),
\end{aligned}
$$

as desired. The third and fourth lines are obtained by direct application of the chain rule of mutual information [32]. All mutual information terms without $\boldsymbol{Z}$ are held constant in the fully offline scenario.

$I(\boldsymbol{Z}; \boldsymbol{M}) \leq I(\boldsymbol{Z}; \boldsymbol{X})$ is a direct consequence of the Data Processing Inequality [32] and the Markov chain $\boldsymbol{M} \to \boldsymbol{X} \to \boldsymbol{Z}$ in Definition 2.2. We now prove the three claims regarding FOCAL, CORRO and CSRO:

1. $\mathcal{L}_{\text{FOCAL}} \equiv -I(\boldsymbol{Z}; \boldsymbol{X})$.

$$
\begin{aligned}
I(\boldsymbol{Z}; \boldsymbol{X}) &:= \mathbb{E}_{\boldsymbol{x}, \boldsymbol{z}} \left[ \log \left( \frac{p(\boldsymbol{z}, \boldsymbol{x})}{p(\boldsymbol{z}) p(\boldsymbol{x})} \right) \right] \\
&= \mathbb{E}_{\boldsymbol{x}, \boldsymbol{z}} \left[ \log \left( \frac{1}{\frac{p(\boldsymbol{z})}{p(\boldsymbol{z}|\boldsymbol{x})} |\mathcal{M}|} \right) \right] + \log(|\mathcal{M}|) \\
&= \mathbb{E}_{\boldsymbol{x}, \boldsymbol{z}} \left[ \log \left( \frac{1}{\frac{p(\boldsymbol{z})}{p(\boldsymbol{z}|\boldsymbol{x})} |\mathcal{M}| \mathbb{E}_{\boldsymbol{x}} \left[ \frac{p(\boldsymbol{z}|\boldsymbol{x})}{p(\boldsymbol{z})} \right]} \right) \right] + \log(|\mathcal{M}|) \\
&\approx \mathbb{E}_{\boldsymbol{x}, \boldsymbol{z}} \left[ \log \left( \frac{1}{\frac{p(\boldsymbol{z})}{p(\boldsymbol{z}|\boldsymbol{x})} \sum_{M^i \in \mathcal{M}} \mathbb{E}_{\boldsymbol{x}^i \sim X^i} \left[ \frac{p(\boldsymbol{z}|\boldsymbol{x}^i)}{p(\boldsymbol{z})} \right]} \right) \right] + \log(|\mathcal{M}|) \\
&= \mathbb{E}_{\boldsymbol{x}, \boldsymbol{z}} \left[ \log \left( \frac{\frac{p(\boldsymbol{z}|\boldsymbol{x})}{p(\boldsymbol{z})}}{\sum_{M^i \in \mathcal{M}} \mathbb{E}_{\boldsymbol{x}^i \sim X^i} \left[ \frac{p(\boldsymbol{z}|\boldsymbol{x}^i)}{p(\boldsymbol{z})} \right]} \right) \right] + \log(|\mathcal{M}|) \\
&= \mathbb{E}_{\boldsymbol{x}, \boldsymbol{z}} \left[ \log \left( \frac{h(\boldsymbol{x}, \boldsymbol{z})}{\sum_{M^i \in \mathcal{M}} h(\boldsymbol{x}^i, \boldsymbol{z})} \right) \right] + \log(|\mathcal{M}|)
\end{aligned}
$$

The first term on RHS is precisely the supervised contrastive learning objective introduced by a variant of FOCAL [22], which is equivalent to the negative distance metric learning loss $\mathcal{L}_{\text{FOCAL}}$ with the effect of pushing away embeddings of different tasks while pulling together those from the same task. Therefore we have

$$I(\boldsymbol{Z}; \boldsymbol{X}) = \mathbb{E}_{\boldsymbol{x},\boldsymbol{z}} \left[ \log \left( \frac{h(\boldsymbol{x}, \boldsymbol{z})}{\sum_{M^i \in \mathcal{M}} h(\boldsymbol{x}^i, \boldsymbol{z})} \right) \right] + \text{const}$$

$$\equiv -\mathcal{L}_{\text{FOCAL}}.$$

2. $\mathcal{L}_{\text{CORRO}} \equiv -I(\boldsymbol{Z}; \boldsymbol{X}_t | \boldsymbol{X}_b)$.

$$I(\boldsymbol{Z}; \boldsymbol{X_t} | \boldsymbol{X_b}) := \mathbb{E}_{\boldsymbol{x}_t, \boldsymbol{x}_b, \boldsymbol{z}} \left[ \log \left( \frac{p(\boldsymbol{z}, \boldsymbol{x_t} | \boldsymbol{x_b})}{p(\boldsymbol{z} | \boldsymbol{x_b}) p(\boldsymbol{x_t} | \boldsymbol{x_b})} \right) \right]$$

$$= \mathbb{E}_{\boldsymbol{x}_t, \boldsymbol{x}_b, \boldsymbol{z}} \left[ \log \left( \frac{1}{\frac{p(\boldsymbol{z} | \boldsymbol{x_b})}{p(\boldsymbol{z} | \boldsymbol{x_t}, \boldsymbol{x_b})} |\mathcal{M}|} \right) \right] + \log(|\mathcal{M}|)$$

$$= \mathbb{E}_{\boldsymbol{x}_t, \boldsymbol{x}_b, \boldsymbol{z}} \left[ \log \left( \frac{1}{\frac{p(\boldsymbol{z} | \boldsymbol{x_b})}{p(\boldsymbol{z} | \boldsymbol{X})} |\mathcal{M}| \mathbb{E}_{M^* \in \mathcal{M}} \left[ \frac{p(\boldsymbol{z} | \boldsymbol{x_t^*}, \boldsymbol{x_b})}{p(\boldsymbol{z} | \boldsymbol{x_b})} \right]} \right) \right] + \log(|\mathcal{M}|)$$

$$\approx \mathbb{E}_{\boldsymbol{x}_t, \boldsymbol{x}_b, \boldsymbol{z}} \left[ \log \left( \frac{1}{\frac{p(\boldsymbol{z} | \boldsymbol{x_b})}{p(\boldsymbol{z} | \boldsymbol{x_t}, \boldsymbol{x_b})} \sum_{M^* \in \mathcal{M}} \frac{p(\boldsymbol{z} | \boldsymbol{x_t^*}, \boldsymbol{x_b})}{p(\boldsymbol{z} | \boldsymbol{x_b})}} \right) \right] + \log(|\mathcal{M}|)$$

$$= \mathbb{E}_{\boldsymbol{x}_t, \boldsymbol{x}_b, \boldsymbol{z}} \left[ \log \left( \frac{\frac{p(\boldsymbol{z} | \boldsymbol{x_t}, \boldsymbol{x_b})}{p(\boldsymbol{z} | \boldsymbol{x_b})}}{\sum_{M^* \in \mathcal{M}} \frac{p(\boldsymbol{z} | \boldsymbol{x_t^*}, \boldsymbol{x_b})}{p(\boldsymbol{z} | \boldsymbol{x_b})}} \right) \right] + \log(|\mathcal{M}|)$$

$$= \mathbb{E}_{\boldsymbol{x}_t, \boldsymbol{x}_b, \boldsymbol{z}} \left[ \log \left( \frac{\frac{p(\boldsymbol{z} | \boldsymbol{x_t}, \boldsymbol{x_b})}{p(\boldsymbol{z})}}{\sum_{M^* \in \mathcal{M}} \frac{p(\boldsymbol{z} | \boldsymbol{x_t^*}, \boldsymbol{x_b})}{p(\boldsymbol{z})}} \right) \right] + \log(|\mathcal{M}|)$$

$$= \mathbb{E}_{\boldsymbol{x}, \boldsymbol{z}} \left[ \log \left( \frac{h(\boldsymbol{x}, \boldsymbol{z})}{\sum_{M^* \in \mathcal{M}} h(\boldsymbol{x}^*, \boldsymbol{z})} \right) \right] + \log(|\mathcal{M}|)$$

$$\equiv -\mathcal{L}_{\text{CORRO}}.$$

3. $\mathcal{L}_{\text{CSRO}} \geq (\lambda - 1) I(\boldsymbol{Z}; \boldsymbol{X}) - \lambda I(\boldsymbol{Z}; \boldsymbol{X}_t | \boldsymbol{X}_b)$.

The CLUB loss in Eqn 6 operates as a upper bound of $I(\boldsymbol{Z}; \boldsymbol{X}_b)$ [31]. By Eqn 5 we have

$$\mathcal{L}_{\text{CSRO}} = \mathcal{L}_{\text{FOCAL}} + \lambda L_{\text{CLUB}}$$
$$\geq \mathcal{L}_{\text{FOCAL}} + \lambda I(\boldsymbol{Z}; \boldsymbol{X}_b)$$
$$= \mathcal{L}_{\text{FOCAL}} + \lambda \underbrace{(I(\boldsymbol{Z}; \boldsymbol{X}) - I(\boldsymbol{Z}; \boldsymbol{X}_t | \boldsymbol{X}_b))}_{\text{chain rule of mutual information}}$$
$$\overset{1}{=} -I(\boldsymbol{Z}; \boldsymbol{X}) + \lambda(I(\boldsymbol{Z}; \boldsymbol{X}) - I(\boldsymbol{Z}; \boldsymbol{X}_t | \boldsymbol{X}_b))$$
$$= (\lambda - 1) I(\boldsymbol{Z}; \boldsymbol{X}) - \lambda I(\boldsymbol{Z}; \boldsymbol{X}_t | \boldsymbol{X}_b).$$

$\square$

### B.2  Proof of Theorem 2.4

*Proof.* Denote by $n_Z := \sum_{i=1}^{n_M} |M^i| |X^i|$ the number of context samples, we have

$$I(\boldsymbol{Z}; \boldsymbol{M}) = H(\boldsymbol{Z}) - H(\boldsymbol{Z} | \boldsymbol{M}) \tag{17}$$

$$\simeq -\sum_{i=1}^{n_Z} \log p(\boldsymbol{z}_i) - \mathbb{E}_M [H(\boldsymbol{Z} | \boldsymbol{M} = M)] \tag{18}$$

$$\simeq -\sum_{i=1}^{n_Z} \log p(\boldsymbol{z}_i) - \sum_{j=1}^{n_M} H(\boldsymbol{Z} | \boldsymbol{M} = M^j). \tag{19}$$

Since $n_Z \gg n_M$, the concentration characteristic of $I(\boldsymbol{Z}; \boldsymbol{M})$ is dominated by the second term. If we ignore the approximation error of the first term, by the Chebyshev's inequality, for any $\epsilon > 0$,

$$\Pr\left(\left|\hat{I}(\boldsymbol{Z}; \boldsymbol{M}) - \bar{I}(\boldsymbol{Z}; \boldsymbol{M})\right| \geq \epsilon\right) \leq \frac{\mathrm{Var}(H(\boldsymbol{Z}|\boldsymbol{M}))}{n_M \epsilon^2}. \tag{20}$$

Or equivalently, with probability at least $1 - \delta$,

$$\left|\hat{I}(\boldsymbol{Z}; \boldsymbol{M}) - \bar{I}(\boldsymbol{Z}; \boldsymbol{M})\right| \leq \sqrt{\frac{\mathrm{Var}(H(\boldsymbol{Z}|\boldsymbol{M}))}{n_M \delta}}. \tag{21}$$

**Remark:** Consider a practical implementation of $p(\boldsymbol{z}|M)$ as a multivariate Gaussian $\mathcal{N}(\boldsymbol{\mu}_M, \boldsymbol{\Sigma}_M)$, which gives $H(\boldsymbol{Z}|\boldsymbol{M}) = \frac{1}{2}\log[(2\pi e)^{\dim Z}|\boldsymbol{\Sigma}_M|]$ [32]. Substituting into Eqn 21, we have with probability at least $1 - \delta$,

$$\left|\hat{I}(\boldsymbol{Z}; \boldsymbol{M}) - \bar{I}(\boldsymbol{Z}; \boldsymbol{M})\right| \leq \sqrt{\frac{\mathrm{Var}(\log|\boldsymbol{\Sigma}_M|)}{2n_M \delta}}. \tag{22}$$

$\square$

## C  Further Experiments

### C.1  UNICORN-SS-0: A Label-free Version

When $\alpha = 0$, UNICORN-SS task representation learning reduces to the minimization of the reconstruction loss only, which becomes a *label-free* algorithm (i.e., it does not require the knowledge of task identities/labels to optimize, as opposed to the contrastive learning in previous methods). We name this special case UNICORN-SS-0, which is equivalent to a concurrent method GENTLE [41]. For a fair comparison, we use a variant of a Bayesian OMRL method BOReL [14] that does not utilize oracle reward functions, as a label-free baseline. At test time, we modify BOReL to use offline datasets rather than the online data collected by interacting with the environment to infer the task information[2]. As shown in Table 5, UNICORN-SS-0 is also competitive with BOReL.

Table 5: UNICORN-SS-0 compared to another label-free COMRL method, BOReL, on Ant-Dir.

| Algorithm | Ant-Dir | |
|---|---|---|
| | IID | OOD |
| UNICORN-SS-0 | 220±16 | 200±9 |
| BOReL | 206±18 | 187±10 |

To further validate that the KL constraint $D_{KL}(q_\theta(z|x)|\mathcal{N}(0, I))$ is unnecessary, we weight the KL constraint to UNICORN-SS-0. Table 6 shows that KL constraint would reduce the performance under the offline setting.

### C.2  Visualization of Task Embeddings

For better interpretation, we visualize the task representations by 2D projection of the embedding vectors via t-SNE [55]. For each test task, we generate a trajectory using each policy in $\{\pi_\beta^{i,t}\}_{i,t}$ and infer their task representations $\boldsymbol{z}$. This setting aligns with our OOD experiments in the main text.

Since the UNICORN-SS objective operates as a convex combination of the reconstruction loss (UNICORN-SS-0) and the FOCAL loss, we compare representations of UNICORN-SS to these

---

[2]Experiment shows that the usage of offline dataset rather than online data has little effect on the results of BOReL.

Table 6: UNICORN-SS-0 compared to other KL constraint weight variants.

| Algorithm | Ant-Dir | |
| --- | --- | --- |
| | IID | OOD |
| UNICORN-SS-0 | 220±16 | 200±9 |
| KL-0.1 | 215±11 | 189±12 |
| KL-0.5 | 202±17 | 182±13 |
| KL-1.0 | 194±11 | 161±7 |
| KL-5.0 | 162±14 | 143±10 |

two extremes. As shown in Figure 6, compared to UNICORN-SS-0, UNICORN-SS provides evidently more distinguishable task representations for OOD data, emphasizing the necessity of joint optimization with the FOCAL loss. On the other hand, despite the seemingly higher quality of task representations of FOCAL, the performance of FOCAL is much worse than UNICORN-SS. We speculate that since FOCAL blindly separates embeddings from different tasks, it may fail to capture shared structure between similar tasks, whereas UNICORN-SS is able to do so by generative modeling via the reconstruction loss.

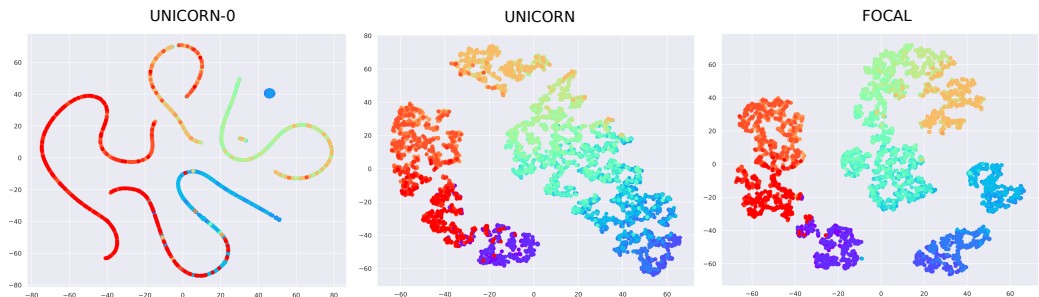

Figure 6: The 2D projection of the learned task representation space in Ant-Dir. Points are uniformly sampled from out-of-distribution data. Tasks of given goals from 0 to 6 are mapped to rainbow colors, ranging from purple to red.

## C.3 More on UNICORN-DT

We provide the learning curves of the results in Section 4.1 here. As shown in Figure 7, UNICORN-SS-DT and UNICORN-SUP-DT demonstrate a faster convergence speed compared to vanilla UNICORN-SS and a higher asymptotic performance compared to vanilla Prompt-DT.

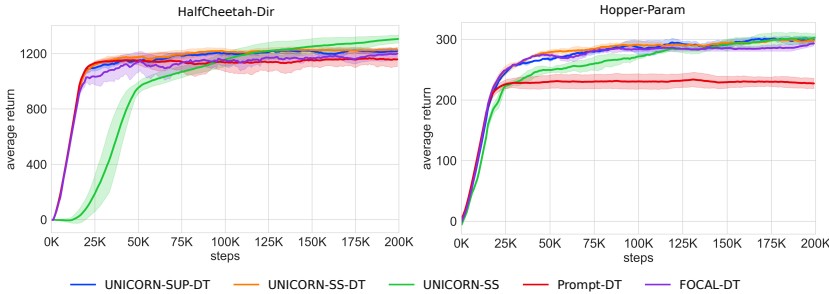

Figure 7: Test returns of UNICORN-SUP-DT and UNICORN-SS-DT against UNICORN-SS, Prompt-DT and FOCAL-DT on 2 benchmarks. The learning curve is averaged by 6 random seeds.

## C.4 Ablation Study

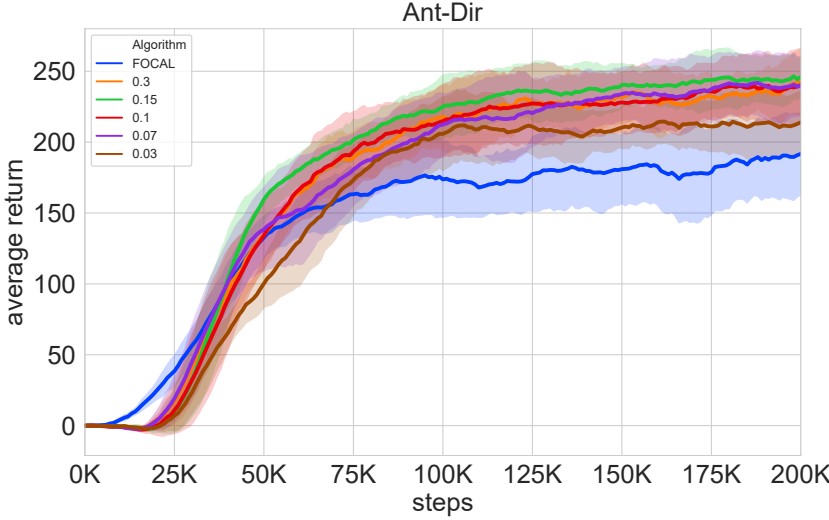

Figure 8: Different hyper-parameter settings of $\frac{\alpha}{1-\alpha}$ on Ant-Dir. The learning curve is averaged by 6 random seeds.

To illustrate the effect of hyper-parameter $\frac{\alpha}{1-\alpha}$ on the asymptotic performance, we set the following ablation study. Figure 8 shows that as $\frac{\alpha}{1-\alpha}$ increases, the performance gradually increases but when it goes excessively the performance would decrease (FOCAL means $\frac{\alpha}{1-\alpha} \to \infty$), which validates our proposed theory. We also find that UNICORN-SS can maintain relatively stable performance over a range of $\frac{\alpha}{1-\alpha}$ values, thus alleviating the exhaustion from parameter-tuning.

## D  Experimental Details

Table 7 lists the necessary hyper-parameters that we used to produce the experimental results.

Table 7: Configurations and hyper-parameters used in the training process.

| Configurations | Ant-Dir | HalfCheetah-Dir | HalfCheetah-Vel | Hopper-Param | Walker-Param | Reach |
|---|---|---|---|---|---|---|
| dataset size | 1e5 | 2e5 | 2e5 | 3e5 | 4.5e5 | 1e5 |
| task representation dimension | 5 | 5 | 5 | 40 | 32 | 2 |
| weight $\frac{\alpha}{1-\alpha}$ | 0.15 | 0.15 | 0.15 | 1.5 | 1.5 | 0.3 |
| training steps | 200k | | | | | |
| task batch size | 16 | | | | | |
| RL batch size | 256 | | | | | |
| context training size | 1 trajectory (200 steps) | | | | | 1 trajectory (500 steps) |
| learning rate | 3e-4 | | | | | |
| RL network width | 256 | | | | | |
| RL network depth | 3 | | | | | |
| encoder width | 64 | | 128 | | | 64 |

