# OpenReview forum: "Towards an Information Theoretic Framework of Context-Based Offline Meta-Reinforcement Learning"
_NeurIPS.cc/2024/Conference — NeurIPS 2024 spotlight_

### Official Review · Reviewer_AbCD · 2024-06-27

**Soundness:** 2
**Presentation:** 2
**Contribution:** 3
**Rating:** 7
**Confidence:** 3

**Summary:**

This paper provides an analysis of existing Offline Meta-RL (OMRL) methods by considering which informational quantity is maximized by which approach, and how they are related.
The show, under a certain assumption on data generation process, that early approaches like FOCAL maximize an upper-bound of the quantity of interest and propose a new method that instead maximizes a lower bound, resulting in a better performance.
They also provide an exhaustive evaluation of their approach in multiple different settings (in-domain, ood, model-based, varying data quality).

**Strengths:**

* The information theoretical analysis seems novel, especially in its analysis and comparison of the prior existing methods.
* The new proposed methods are well motivated based on the prior analysis and perform well on a diverse set of problem setting variations
* Soure code is provided

**Weaknesses:**

* The presentation of the Information Theoretical Framework in Section 2.2 is kind of hard to follow. The distinction of "minimally causally related" and "maximally causally related" quantities seems nonstandard to me and should be clarified. Maybe just writing $I(Z;M) \geq I(Z:M|X_b)$ is more clear? I would also appreciate a more clear justification of this assumption.

 * Experimental validation explores a broad set of settings, but does not discuss hyper-parameter choice and optimization, except for one ablation. A proper hyper-parameter optimization for baselines and proposed methods would significantly strengthen the reliability of the evaluation.

 * Theorem 2.4 (Concentration Bound for Supervised UNICORN) is just a straight-forward application of Chebyshev, is not discussed further and doesn't seem to provide any deeper insights. It is thus unclear to me why it was included in the paper, what additional insights it provides.

Minor Points:
 * Figure 1 a) and 1 b) seem unnecessarily complex, a more sketch-like format might be easier to understand
 * It is not clear from the paper how exactly the "Supervised" baseline encodes the context.
 * Interaction Information $I(X;Y;Z)$ is not commonly used outside the causal inference community as far as I know, so it would be nice to quickly define it for the reader.

**Questions:**

* As mentioned in the weaknesses above, my biggest concern is the interpretation of assumption 2.2, i.e.  $I(Z;M) \geq I(Z:M|X_b)$. Some additional justification or clarification would be helpful. When does it apply? When is it violated?

* It would be nice to include ContraBAR [Choshen and Tamar; ICML2023] into the comparison of methods, as I believe it also maximizes $I(Z;M)$ in this framework, and is also predictive (by using Contrastive Predictive Coding). It also is applied online and offline. Without it, this submission seems to suggest that it provides the first method maximizing $I(Z;M)$, which I don't believe is correct.

* What insights does Theorem 2.4 provide?

* Are the proposed methods limited to discrete task spaces (during training and execution?)? If so, how could the proposed methods be extended to continuous task spaces?

I'm willing to improve my score if the concerns, especially about Assumption 2.2, are addressed.

**Limitations:**

* The methods as presented seem to be limited to discrete sets of tasks. Adapting them to continuous state should be feasible, for example by using InfoNCE in UNICORN-Sup, but as such variants are not discussed or evaluated this limitation should be stated.

At least it should be mentioned in 3.1) when introducing the evaluation tasks.

---

> ### Author Rebuttal · Authors · 2024-08-07
>
> We sincerely thank the reviewer for the constructive comments, which really help us make the paper stronger. Our response to your concerns/questions:
>
> # Justification of Assumption 2.2
>
> For clarification of $I(Z; M; X_b) \ge 0$ or equivalently $I(Z; M) \ge I(Z; M|X_b)$, please see our following response to Q1. "minimally causally related" and "maximally causally related" are indeed qualitative descriptions to convey that the causality between $Z$ and $X_t$ is stronger compared to that between $Z$ and $X_b$. According to the three levels of causation [1], we can define the former as a conditional causality, which means $X_t$ is a $\textit{necessary}$ factor to infer task identity $Z$, but not sufficient. When conditioning on $X_b$, it provides sufficient information about the reward and transition function, which makes the corresponding $I(Z; X_t|X_b)$ the primary causality.
>
> In constrast, the causal relation between $Z$ and $X_b$ is much weaker, since the distribution of $X_b$ mostly depends on the behavior policy $\pi_{\beta}$, a confounding factor. Therefore the relation should be considered a contributory causality, which means the knowledge about $X_b$ may provide some information about $Z$, but is neither necessary nor sufficient to fully determine $Z$, which induces our definition of $I(Z; X_b)$ as the lesser causality. It's worth to note that quantifying causality in terms of mutual information which we did in our paper is also commonly seen in other literature [2] [3].
>
> # Hyper-Parameters
> We set the same batch-size, learning rate and network for all context-based methods. The same principle applies for all gradient-based methods and transformer-based method. Additionally, by adjusting the head of the transformer, we ensure that its number of parameters stay close to the policy network in gradient-based method and context-based methods.
>
> For all methods on each environment, we do thorough grid-search to select the hyper-parameters:
>
> Prompt-DT: $K~[2,5,10,40]$, same as its paper.
>
> MACAW: $\lambda\sim [0.005, 0.01, 0.05, 0.1]$, where 0.01 is recommended in its paper.
>
> FOCAL: $\beta\sim[1,2,8,16]$ and $n\sim[-2,-1,1,2]$, same as its paper.
>
> CORRO: we fine-tune the training process of the negative-generator, reducing the training loss from 2.833 (open-source repo) to 0.084.
>
> CSRO: $\lambda_{FOCAL}\sim[0.01, 0.03, 0.07, 0.1, 0.3, 0.7, 1.5, 3, 7, 10, 25, 50]$ and $\lambda_{CLUB}\sim[0.01, 0.03, 0.07, 0.1, 0.3, 0.7, 1.5, 3, 7, 10, 25, 50]$, where $10,25,50$ are recommended in its paper.
>
> UNICORN-SUP: no hyper-parameter to tune. Hence, UNICORN-SUP is recommended if no hyper-parameter tuning is a priority.
>
> UNICORN-SS: $\frac{\alpha}{1-\alpha}\sim [0.01, 0.03, 0.07, 0.1, 0.3, 0.7, 1.5, 3, 7, 10, 25, 50]$. Despite our search for UNICORN-SS, we find that it achieves better performance on many parameter choices.
>
> Due to the space limit, we demonstrate the HP records of Prompt-DT, MACAW and CSRO on Ant-Dir in our general response PDF Figure 2.
>
> # Thm 2.4
>
> Please refer to our following response to Q3.
>
> # How UNICORN-SUP encodes the context
>
> We use the output feature of the encoder as $z$, which is also the input of the classifier.
>
> # Questions
>
> Q1. "Interpretation of $I(Z; M)\ge I(Z; M|X_b)$"
>
> We now provide a rigorous proof:
>
> $I(Z; M|X_t, X_b) = 0 \Longrightarrow I(Z; M) - I(Z; M|X_t, X_b) = I(M; X_b, X_t) - I(M; X_b, X_t|Z) \ge 0$
>
> $\Longrightarrow I(M; X_b) + I(M; X_t|X_b) - I(M; X_b|Z) - I(M; X_t|X_b, Z) \ge 0$
>
> $\Longrightarrow I(M; X_b) - I(M; X_b|Z) \ge \underbrace{-I(M; X_t|X_b)}_{const} + \underset{\ge 0}{I(M; X_t|X_b, Z)} \ge const$
> .
> Since $I(Z; M) - I(Z; M|X_b) =  I(M; X_b) - I(M; X_b|Z)$, we have $I(Z; M) - I(Z; M|X_b) \ge const$
>
> Namely, for offline setting, $I(Z; M)\ge I(Z; M|X_b)$ holds up to a constant, which can directly be used to prove the LHS of Thm 2.3 (line 485). We are grateful that your question helps us rethink and rigorize our theory. With the proof above, Assumption 2.2 is no longer an assumption, we will make it a definition of graphical models of COMRL instead in the final version.
>
> Q2. ContraBAR
>
> Thank you for the advice. ContraBAR is indeed an interesting paper, which considers contrastive learning for Bayes-Adaptive RL. Please check the pdf in our general reponse for empirical comparison between UNICORN, ContraBAR and BOReL. UNICORN still maintains its edge.
>
> Based on Eqn 3 of the paper, contraBAR optimizes $I(Z; X_t|A)$, which closely resembles the primary causality $I(Z; X_t|X_b) = I(Z; X_t|S, A)$ in our paper and the CORRO loss, but not $I(Z; M)$ directly. Also it follows the paradigm of VariBAD and BOReL which aggregates a long history/trajectory to infer the belief $z$ (therefore non-Markovian), considers online adaptation at test time; in contrast, UNICORN uses single-step transition for task inference (Markovian), and should be applicable to both offline and online adaptation since it's robust to behavior OOD.
>
> Q3. Thm 2.4
>
> Please see our general response Q4.
>
> Q4.  discrete task spaces
>
> The 6 MuJoCo and MetaWorld benchmarks used in the current paper are all continuous control environments, so it has not been tested on discrete task spaces. We agree with this limitation. However, extending UNICORN to discrete space should be straightforward: for task representation learning, replace the model backbone to one that can deal with discrete observation, e.g., CNN/ViT for image and transformer/LSTM for language tokens; for downstream RL, replace SAC agent with its variant that can handle discrete actions [4] or other policy networks such as DQN [5]. Such experiments can be addded to the 1 extra page should the paper be accepted.
>
> [1] http://media.acc.qcc.cuny.edu/faculty/volchok/causalMR/CausalMR3.html
>
> [2] Conditional Mutual Information to Approximate Causality for Multivariate Physiological Time Series
>
> [3] On Causality and Mutual Information
>
> [4] Soft actor-critic for discrete action settings
>
> [5] Human-level control through deep reinforcement learning

---

> ### Comment · Reviewer_AbCD · 2024-08-08
>
> Thank you for the rebuttal. I'm a bit busy currently so I will go through the other parts later, but concerning the limitation:
>
> > Q4. discrete task spaces
> > The 6 MuJoCo and MetaWorld benchmarks used in the current paper are all continuous control environments, so it has not been tested on discrete task spaces. We agree with this limitation. However, extending UNICORN to discrete space should be straightforward: for task representation learning, replace the model backbone to one that can deal with discrete observation, e.g., CNN/ViT for image and transformer/LSTM for language tokens; for downstream RL, replace SAC agent with its variant that can handle discrete actions [4] or other policy networks such as DQN [5]. Such experiments can be addded to the 1 extra page should the paper be accepted.
>
> This seems to be a misunderstanding. I meant that the approach seems to be limited to discrete sets of hidden-parameters, rather than a continuous hidden parameters. I mistyped in my original review, it should be "Adapting them to continuous hidden states should be feasible".
>
> Let's take the examle of Ant-Dir in Figure 1. Intuitively we would like a method that is able to go to any goal position on the circle, not just to the ~20 test goals.
> An extension to such a setting would probably be nontrivial and it would be nice to discuss potential ways of doing so in the paper, or at least mentioning this limitation.

---

> ### Author Response · Authors · 2024-08-09
> **Rebuttal**
>
> Thank you for your clarification. There seems to be a misunderstanding about how the context encoder and task representation $Z$ work in UNICORN. Same as FOCAL and other related baselines, our encoder takes a transition tuple $x=(s_t, a_t, r_t, s_{t+1})$ as input and outputs a representation vector $z$ in a $\textit{continuous}$ latent space. Please refer to Figure 6 for visualization of these projected $\textit{continuous}$ representations. At test time, the meta policy performs few-shot generalization by taking new context $X'$ from the test task $M'$, and infer its new representation $Z'$ the same way to condition the downstream RL networks.
>
> In the example of Ant-Dir, the context encoder generates a $\textit{distinct}$ task representation/statistic for each individual test goal. Therefore, for any goal position on the circle, as long as we are given its context, the context encoder can generate a specific $Z$ for conditioning the RL agent to solve the corresponding task (i.e., navigating to the specific goal position). The specific set of 20 test goals in Figure 1 is only used to empirically demonstrate the generalization of the learned task representation as well as the conditional RL networks. We believe our algorithm should have no problem generalizing to $\textit{any}$ test task sampled from the training distribution (based on result in Sec 3). For out-of-distribution tasks, which is discussed in Figure 1 c) and Sec 4.2, it's indeed extremely challenging, but the generator of UNICORN-SS can still provide nontrivial improvement compared to other baselines, shown in Figure 5.
>
> Again thank you very much for your additional comments. Please let us know if your concern is properly addressed and if there is any misunderstanding or further questions.

---

> > ### Comment · Reviewer_AbCD · 2024-08-09
> >
> > Thank you for the reply. I appeciate the clarification, but I made a mistake in my prior comment.
> >
> > The limitation I was interested in is **during training**.
> >
> > Supervised Unicorn is using an N-way classification loss. This probably works very well with discrete sets of tasks, i.e. the N=20 training tasks sampled from the semicircle in Figure 1. If I'm interpreting the Appendix correctly, you have 200 transitions for each of those tasks? ("Context training size in Table 7) This means 4000 transitions total.
> >
> > However, if we instead sample N=4000 tasks from the semicircle, with one transition each, I would assume that training a classifier becomes problematic. This is what would happen in a continuous task space, and in this sense, the method probably does not work in continuous task spaces.
> >
> > However, as the authors pointed out in the previous reply, it can generalize from a discrete set of training tasks, so my concerns are resolved about this matter.
> >
> >
> > I'm also happy with the remainder of the rebuttal and after reading it and the other reviews I will increase my score from 6->7.

---

> ### Author Response · Authors · 2024-08-10
> **Further Clarification**
>
> Thank you for the comments. The "context training size" refers to the number of transitions used for each task during $\textit{one forward pass}$ (basically we randomly sampe 1 trajectory for task inference). The total number of transitions used to train a single task is given by the "dataset size in Table 7", and is much larger. For example, for Ant-Dir, it's $1e5=100,000$. We will clarify this in the final version.
>
> We agree that if we increase the number of tasks without expanding the dataset proportionally, it might be problematic not just for task representation learning, but also for downstream offline RL since it may suffer from insufficient dataset coverage [1] for each task. However, the hope is that, just like pretraining large language models, we are able to collect decent amount of data for each individual task, and use offline meta-RL as a multi-task pretraining technique to produce robust task representations for a generalist agent to solve decision making problems. This is why we are interested in DT since transformer is a promising backbone with better scaling in the large data regime.
>
> Again thank you very much for the fruitful discussion. Please do not hesitate to let us know if you have further comments/questions.
>
> [1] What are the Statistical Limits of Offline RL with Linear Function Approximation?

---

### Official Review · Reviewer_7Lx7 · 2024-07-11

**Soundness:** 3
**Presentation:** 3
**Contribution:** 3
**Rating:** 7
**Confidence:** 3

**Summary:**

In this paper, the authors propose a unified mathematical framework that encapsulates a subset of the developments conducted in Context-Based Offline Meta-Reinforcement Learning. They achieve this by carefully describing each previous attempt to solve this problem, and how the incremental improvements have been conducted by building upon previously proposed models. Next, they note that all of these previous works can be described as different but related information theory quantities, largely corresponding to the bounds of the (proposed) quantity that should ideally be optimally maximized, corresponding to the mutual information between the latent variables and task variables. They thoroughly explain each component of the theory framework by performing mathematical analysis and giving graph descriptions that enhance comprehension. Based on this framework, the authors propose a method called UNICORN, designed to maximize the ideal mutual information, for supervised and self-supervised settings. Then, they perform experiments under different conditions comparing the proposed method to the previous attempts described in the paper, achieving better results across various experimental settings.

**Strengths:**

- This paper is very well motivated by highlighting previous work contributions to the field and how they fit with their framework.
- The paper strikes a good balance between exposing technical details and explaining the intuition behind each component of the mathematical analysis. By clearly linking each component to previous work, the authors effectively unify the goals of the models previously mentioned in the paper.
- By indicating what is the ideal quantity to be maximized, they propose two loss functions, one for supervised and one for self-supervised learning, also explaining how this can be understood in light of previous work.
- The experiments are well-designed, comparing UNICORN to the relevant and up-to-date models encompassed by its framework. Additionally, UNICORN achieves better results than the baseline while being grounded in a more principled methodology.

**Weaknesses:**

- The clarity of some aspects of the mathematical derivations and theoretical conclusions could be improved (see specific questions below).

- This is likely a matter of presentation rather than a fundamental issue the approach.
The paper would benefit from a clearer explanation of which relevant works from the existing literature are not explicitly incorporated into the proposed framework (see specific questions below).

**Questions:**

1.- In equation 7, what happened to the term $H(M)$? The only explanation I found that makes it disappear is that if you are maximizing mutual information, that term is irrelevant for the maximization since it's fixed, but that doesn’t mean $I(Z;M) = -H(M|Z)$?

2.- The paragraph starting at line 156 discusses CSRO optimizing a convex interpolation of the upper bound $I(Z;X)$ and lower bound $I(Z;X_t, | X_b)$. How is this beneficial compared to just optimizing the lower bound? If the upper bound is
$$ I(Z;X) = I(Z; X_{t}|X_{b}) + I(Z;X_{b}), $$ then the convex combination of the upper and lower bound would be $$ I(Z; X_{t}|X_{b}) + \lambda I(Z;X_{b}) $$(also similar to the self supervised objective in eq 12), and according to the framework, the second term includes correlations due to the current policy and not the model of the environment. How optimizing this convex combination would be beneficial? In practice seems like this is a good idea based on the performances of UNICORN and CSRO compared to FOCAL and CORRO.

3.- Are there other methods that try to solve COMRL that are not captured by your framework? What are the differences in their formulation and why can’t they be captured by your framework, and how you would potentially go about integrating these to the theory?

4.- Can this be extended to Online Meta RL? at least from the formalization seems that it could be directly applicable, except for the time-evolving policy which can make $I(Z;X_{b})$ difficult to deal with.

**Limitations:**

The authors discuss some limitations, and the questions above may address potentially new ones.

---

> ### Author Rebuttal · Authors · 2024-08-07
>
> We sincerely thank the reviewer for the constructive comments and the appreciation of our work. Our response to your concerns/questions:
>
> Q1. Eqn 7
>
> Eqn 7 concludes that $I(Z; M)\equiv -H(M|Z)$, which by definition of equivalence $\equiv$ in Thm 2.3, suggests that the two terms are equal up to a constant, which is indeed $H(M)$. You are correct, $H(M)$ is dropped because it doesn't involve $Z$, and can therefore be regarded as a constant for optimization.
>
> Q2. "How is convex interpolation beneficial?"
>
> The joint distribution of (s, a) depends on several factors: the initial state distribution $\rho_0$, the transition function $T$ and the behavior policy $\pi_{\beta}$. Therefore even it's highly correlated with $\pi_{\beta}$, it does in principle contain $\textit{some}$ causal information about the task identity $M$, especially for tasks that differ in transition dynamics (e.g., Walker, Hopper). Completely dropping the term might be an overkill. We briefly explained this in line 159-161 but will further clarify in the final version. Please check our general reponse for more discussion.
>
> Q3. Scope of the framework
>
> Very interesting question. The level of integration of existing methods to our theory depends on the level of abstraction. Given our Table 1 which summarizes the most represenative COMRL methods to our knowledge, mathematically, we expect our framework to be able to incorporate almost all existing CORML methods that focus on task representation learning. Since according to our Definition 2.1, as long as the method tries to solve CORML by learning a sufficient statistic $Z$ of $X$ w.r.t $M$, it will eventually come down to optimizing an information-theoretic objective equivalent to $I(Z; M)$, or a lower/upper bound like the ones introduced by Thm 2.3, up to some regularizations or constraints. At this level of abstraction, we cannot think of any exception to our best knowledge but are definitely open to discussion.
>
> Since our framework assumes a decoupling of task representation learning and offline policy optimization, UNICORN currently doesn't account for COMRL methods which make nontrivial modification to the standard actor-critic RL learning loop.  This limitation is discussed in line 341-343. For these methods, however, we expect UNICORN to be able to act as a plug-and-play task representation learning module to further improve their task inference robustness and overall performance.
>
> Moreover, if we look at the implementation level, there could be myriads of design choices even for optimizing the same objective. For example, replacing the VAE generator in UNICORN-SS by a diffusion/GAN model might be helpful for image-based tasks, using DT/Diffusion [1] RL backbones might exhibit better scaling for larger datasets and more complex tasks. Also there can be different paradigms, such as the Bayes-Adaptive RL [2][3] which infer $z$ as a belief of the whole history in pursuit of bayes-optimal control, at the cost of breaking the Markovian property of the MDP as well as higher computation. From this perspective, the current supervised and self-supervised UNICORN formulations are far from a complete coverage of all these possible options. Nevertheless, we believe our framework provides a principled guideline for designing novel CORML algorithms from the task representation learning perspective, and can be further extended and improved on a case-by-case basis (e.g., for specific settings or sub-problems of COMRL).
>
> Q4. Extending UNICORN to Online Meta-RL
>
> We agree that the mathematical formalization should be directly applicable to online RL. However, many of our key derivations rely on the static assumption of $M$ and $X$ (e.g., line 485), which are apparently violated in the online scenario. Extending our framework to online RL is interesting and nontrivial, which might be a good topic for future work.
>
> [1] MetaDiffuser: Diffusion Model as Conditional Planner for Offline Meta-RL
>
> [2] ContraBAR: Contrastive Bayes-Adaptive Deep RL
>
> [3] Varibad: A very good method for bayes-adaptive deep rl via meta-learning

---

> ### Comment · Reviewer_7Lx7 · 2024-08-09
>
> Thanks to the authors for their response to my concerns. Regarding Q1, equation 7 should be corrected or clarified concerning the dropping of the $H(M)$ term to avoid confusion. Also, adding the discussion of Q2, Q3, and Q4 in the paper would be nice, as it opens interesting avenues for future work.
>
> Based on the authors' responses and other reviewers' comments, I will keep my scores and increase my confidence.

---

> > ### Author Response · Authors · 2024-08-09
> > **Thank You**
> >
> > Thank you for your comments and suggestions, which will undoubtedly improve the quality of our paper. We will make sure to properly address your questions in the final version.

---

### Official Review · Reviewer_sHdF · 2024-07-14

**Soundness:** 3
**Presentation:** 4
**Contribution:** 4
**Rating:** 8
**Confidence:** 4

**Summary:**

This work introduces two new algorithms for COMRL based on an information theoretic decomposition of behaviour and environment information. The primary insight is that when encoding the contextual task representation as much environment information should be maintained while as little behaviour information is maintain. The UNICORN-SS and UNICORN-SUP algorithms ultimately build on the FOCAL algorithm with the addition of either a reconstruction or classification loss. In the case of classification loss, a portion of the training network must predict which task is being performed from the inferred latent task representation and behaviour component of the context. In the reconstruction loss, a portion of the network must reconstruct the next state and reward - the task component of the context - from the latent representation and behaviour component. Experiments show that UNICORN-SS and UNICORN-SUP have more consistent performance on in-distribution samples and superior performance on out-distribution samples.

**Strengths:**

# Originality
The decomposition of context into behaviour and task information is interesting and allows for a clear classification of previous COMRL tasks. This more abstract grouping of algorithms is helpful and new.

# Clarity
Overall this work does an excellent job of introducing multiple algorithms and consideration clearly. The main differences between FOCAL, CORRO and CSRO are clear and intuitive from the equations presented. The notation used is intuitive and clear, and the paper is well written. Figures are clear and easily understadible.

# Quality
The aim of the work is clear and the logic and structure of the argument is excellent. Experiments are appropriate for assessing the utility of the new algorithms and the the baselines are challenging and the appropriate choices to my knowledge.

# Significance
The empirical results of this work are convincing for me that this indeed makes a significant contribution to the field. I appreciate the improved consistency on in-distribution performance as the improved out-distribution performance as getting less variance in RL tasks is not easy in its own right. Overall I think this is a very good paper.

**Weaknesses:**

My weaknesses are relatively minor.

# Clarity
Legends on Fig 1 should be larger. There were also a couple statements which were unclear but I ask about these in Question below. The statement in Theorem 2.4 was difficult to understand and I relied on Equation 11 for meaning. I think it could just be a grammar error. As a suggestion I would put the equality $I(Z;X) = I(Z; X_t | X_b) + I(Z;X_b)$ in the inequality of Theorem 2.3.

# Significance
I do think the theorems need proof sketches. This is because those theorems are the main source of new insight from this work - particularly Theorem 2.3. Due to the nature of information theory proving this theorem is likely very familiar and the construction of the COMRL problem to fit the format is the bigger advance, I see that, but it would still be helpful to get some brief intuition for how the result emerges in the main text. It would also be helpful if it was more explicitely stated why $L_{focal}$, $L_{CORRO}$ and $L_{CSRO}$ are grouped into those mutual information types. Looking at their individual equations I can see from the variables present why these types were allocated but the clearer statement on why would be useful. My last weakeness on significance is one the authors acknowledge (which I appreciate already) - due to the broad nature of the mutual information theory representation it is not clear why UNICORN outperforms the other CORRO or another algorithm which operates on the primary causality or a convex combination of causalities. I do not however see how the authors could account for this and so I make this point lightly.

**Questions:**

1. I'm not certain on what is being presented in Fig 1a) with the red line. The message of the figure is clear, but if the red line is showing behaviour of an agent on the task then why would it not be more correlated with the goal direction?
2. The last paragraph of page 4 makes the claim that using a context combination of causalities might explain why CSRO outperforms FOCAL and CORRO. It is not clear to me why including any lesser causality would improve the latent representations. Since this influenced UNICORN I feel this is  important for me to understand
3. In Table 1, column Context  X. What does "Transition" vs "Trajectory" mean?
4. For Equation 14. Why would switching to sampling from the encoder guarantee that the expected log probability will decrease from the line  above? Is there an assumption here or something I am missing?
5. Figure 3. was extremely useful but even still I find that I am unsure exactly of how the context is obtained from the replay buffer. Is it correct that the latent task variable is inferred from a subset of the replay buffer with multiple steps, while the $x$ variables are a single sample which is being trained on? Perhaps annotation the middle and bottom arrows from context as you did with the top branch would be helpful.
6. In Table 3. Surely no algorithm should work for a random policy. By definition the random policy is not goal directed and so it should be impossible to infer task from this dataset with the same state and action space?

**Limitations:**

Limitations are stated and clear.

---

> ### Author Rebuttal · Authors · 2024-08-06
>
> We sincerely thank the reviewer for the constructive comments and the appreciation of our work. Our response to your concerns/questions:
>
> # Clarity
>
> Thank you for your advice. We will make sure to improve the presentation of figures and theorems in the final version.
>
> Theorem 2.4 mainly bounds the error of estimating the real expectation $\bar{I}(Z; M)$ with $n_M$ finite task instances drawn from the real task distribution $p(M)$. Please check our general response for in-depth discussion.
>
> Adding $I(Z; X) = I(Z; X_t|X_b) + I(Z; X_b)$ in Theorem 2.3 is a good idea.
>
> # Significance
>
> - proof sketch:
>
> In the famous InfoNCE paper [1], it is shown that the contrastive loss $L_N$ as in FOCAL [2] upper-bounds the mutual information of original signal $x$ and its representation $z$: $I(Z, X) \ge \log (|M|) - L_N$. If we sample the positive and negative pairs in $L_N$ according to the real task distribution $p(M)$, we can turn this inequality to equality, which proves that $L_{FOCAL} \equiv -I(Z, X)$. For CORRO, it's the same format except that all samples are conditioned on the same $x_b$, which is equivalent to the primary causality $I(Z, X_t|X_b)$. For CSRO, the additional CLUB loss in Eqn 6  provides an upper bound of the lesser causality $I(Z, X_b)$. By combining it with the FOCAL loss ($I(X, Z)$ = $I(Z; X_t|X_b)$ + $I(Z, X_b)$), the CSRO loss is effectively a linear combination of the primary and lesser causalities.
>
> - why UNICORN outperforms baselines:
>
> As explained in our following reponse to Q2, the lesser causality $\textit{does}$ contain some causal information about the task identity, therefore Thm 2.3 establishes the information-theoretic superiority of UNICORN over FOCAL and CORRO, since FOCAL introduces spurious correlation whereas CORRO overlooks part of the causal information. The comparison between UNICORN and CSRO is trikier, since the objective of CSRO and UNICORN-SS are mathematically equivalent. We believe the main difference lies in the implementation choice. CSRO approximates $I(Z; X_b)$ with the CLUB loss. On top of this approximation, it employs variational distribution $q_{\theta}(z|x_b)$ to approximate the intractable real conditional distribution $q(z|x_b)$ in Eqn 6. Therefore it involves $\textit{2 levels}$ of approximation. In contrast, both UNICORN variants involves only ${1 level}$ of approximation (finite sample approximation in Thm 2.4 for UNICORN-SUP and variational approximation in eqn 14 for UNICORN-SS). We believe this explains (at least partially) the superiority of UNICORN over CSRO. Please check our general response for in-depth discussion.
>
> # Questions
>
> Q1. "why would the red line in Fig 1a) not be more correlated with the goal direction?"
>
> The red lines consist of 20 trajectories randomly picked from our mixed-quality dataset (used in sec 3.2, 3.3), which contains trajectories logged throughout the entire training cycle of a behavior policy agent. Therefore it should be a mixture of random, medium and expert-level trajectories, which is why they are not sufficiently correlated with the goal direction.
>
> Q2. "why including any lesser causality would improve the latent representations?"
>
> The joint distribution of (s, a) depends on several factors: the initial state distribution $\rho_0$, the transition function $T$ and the behavior policy $\pi_{\beta}$. Therefore even it's highly correlated with $\pi_{\beta}$, it does in principle contain $\textit{some}$ causal information about the task identity $M$, especially for tasks that differ in transition dynamics (e.g., Walker, Hopper). Completely dropping the term might be an overkill. We briefly explained this in line 159-161 but will further clarify in the final version.
>
> Q3. "Transition" vs "Trajectory"
>
> "Transition" corresponds to the single-step tuple $(s_t, a_t, r_t, s_{t+1})$, whereas "trajectory" stands for a complete history $(s_0, a_0, r_0, s_1, \ldots, s_t)$ of MDP up to a certain time step $t$.
>
> Q4. Last inequality of eqn 14.
>
> Since the true $q(z|x)$ is unavailable, it's a common practice in variational inference [3] to approximate it by the posterior of the encoder $q_{\theta}(z|x)$. The inequality is caused by approximating the true probability $p(x_t|z, x_b)$ with a decoder $q(x_t|z, x_b)$
>
> Proof: let $q$ denote $p_{\theta}$ in eqn 14,
>
> $-I(X_t; Z, X_b) \equiv -\mathbb{E}_{x_t, x_b, z} [\log p(x_t|z, x_b)]$
>
> $= -\mathbb{E}_{x_t, x_b, z}[\log \frac{p(x_t|z, x_b)}{q(x_t|z, x_b)}]$
>
> $-\mathbb{E}_{x_t, x_b, z}[\log q(x_t|z, x_b)]$
>
> $= -\mathbb{E}[KL(p(x_t|z, x_b)||q(x_t|z, x_b))] + L_{\text{recon}} \le L_{\text{recon}}$
>
> Q5. "how the context is obtained from the replay buffer"
>
> For context encoder: we randomly sample one trajectory $\tau$ from the replay buffer, compute $z$ for every single-step transition $x$ in $\tau$, and take the averaged $\bar{z}$ as the representation.
>
> For downstream RL training: we randomly sample a batch of single-step transitions $x$ from the replay buffer for policy optimization.
>
> In principle, one can use the exact same sampling strategy for both context encoder and RL training. Our implementation is meant for easier integration with other baselines. Thank you for your advice, we will make this more clear in the final version.
>
> Q6.  "it should be impossible to infer task from random-quality dataset"
>
> For Ant-Dir, the environments have dense rewards and tasks only differ in reward function. Therefore even with the same state and action distribution, the tasks should still be distinguishable by looking at the reward $r$ conditioning on each $(s, a)$ tuple. Thus the superior performance of UNICORN on random-quality dataset is a strong evidence of its robustness.
>
> [1] Representation learning with contrastive predictive coding
>
> [2] Provably improved context-based offline meta-rl with attention and contrastive learning
>
> [3] Auto-encoding variational bayes

---

> > ### Comment · Reviewer_sHdF · 2024-08-12
> > **Comment by Reviewer sHdF**
> >
> > I thank the authors for their detailed response. Indeed my concerns are generally addressed. I think the agreed points to be clarified will make a difference. I also hope some of the points which I was corrected on above will be used to improve the clarity and avoid similar confusions in a revised draft. I thank the authors for the nice paper and will raise my score to 8.

---

> > > ### Author Response · Authors · 2024-08-12
> > > **Thank You**
> > >
> > > Thank you very much for the constructive comments. We highly appreciate the fruitful discussion, which will undoubtedly improve the quality of our paper. We will make sure to properly implement all reviewers' suggestions in the final version.

---

### Author Rebuttal · Authors · 2024-08-07

We sincerely thank the reviewers, ACs and PCs for ensuring high quality review of the paper. We find all reviews constructive and helpful for making our paper stronger. As requested, the attached PDF provides result of UNICORN vs. a new baseline, ContraBAR, as well as hyper-parameter optimization records of baselines including Prompt-DT, MACAW and CSRO.

Here we summarize some key/common questions raised and provide our general respose as follows:

Q1. Clarification of Assumption 2.2, in particular $I(Z; M; X_b) \ge 0$ or equivalently $I(Z; M) \ge I(Z; M|X_b)$?

We are grateful to reviewer AbCD who helped us rethink and futher rigorize our theory. This assumption was originally proposed as a key step for proving $I(Z; M) \ge I(Z; X_t|X_b)$ (line 485). However, we now conclude that it can be rigorously proven up to a constant, which is sufficient for deriving Thm 2.3. Please check the detailed proof in our response to reviewer AbCD, Q1.

Q2. About Thm 2.3, why is adding the lesser causality $I(Z; X_b)$ beneficial?

The joint distribution of (s, a) depends on several factors, including the initial state distribution $\rho_0$, the transition function $T$ and the behavior policy $\pi_{\beta}$. Therefore it contains two components: the correlation with $\pi_{\beta}$, which is spurious; the correlation with $\rho_0$ and $T$, which are contributory causalities [1] that capture partial information about $M$. Therefore, optimizing the primary causality $I(Z; X_t|X_b)$ alone eliminates the spurious correlation at the cost of overlooking the contributory causalities, which is likely an overkill. Instead, an ideal choice would be a trade-off between the spurious correlation and the contributory causalities, effectively what UNICORN-SS and CSRO do, resulting in better empirical performance.

Q3. Scope of UNICORN: Are there any COMRL methods not captured by the framework? If yes, why and how to incorporate them?

The level of integration of existing methods to our theory depends on the level of abstraction. Given Table 1 which summarizes the most represenative COMRL methods to our knowledge, mathematically, we expect our framework to be able to incorporate almost all existing COMRL methods that focus on task representation learning. Since according to our Definition 2.1, as long as the method tries to solve COMRL by learning a sufficient statistic $Z$ of $X$ w.r.t $M$, it will eventually come down to optimizing an information-theoretic objective equivalent to $I(Z; M)$, or its lower/upper bound like the ones introduced by Thm 2.3, up to some regularizations or constraints. At this level, we are unaware of any exception but are definitely open to discussion.

Since our framework assumes a decoupling of task representation learning and offline policy optimization, UNICORN currently doesn't account for COMRL methods which make nontrivial modification to the standard actor-critic RL learning loop. This limitation is discussed in line 341-343. For these methods, however, we expect UNICORN to be able to act as a plug-and-play task representation learning module to further improve their task inference robustness and overall performance.

Moreover, if we look at the implementation level, there could be myriads of design choices even for optimizing the same objective. For example, replacing the VAE generator in UNICORN-SS by a diffusion/GAN model might be helpful for image-based tasks, using DT/Diffusion [2] backbones might exhibit better scaling for larger datasets and more complex tasks. Also there can be different paradigms, such as the Bayes-Adaptive RL [3][4]. From this perspective, the current supervised and self-supervised UNICORN formulations are far from a complete coverage all these possible choices. Nevertheless, we believe our framework provides a principled guideline for designing novel COMRL algorithms from the task representation learning perspective, and can be further extended and improved on a case-by-case basis (e.g., specific settings or sub-problems of COMRL).

Q4. Significance of Thm 2.4.

The interpretation of Thm 2.4 is only briefly discussed in line 344-345 due to the page limit. The main purpose of Thm 2.4 is to show that, even though UNICORN-SUP seems to directly optimize $I(Z;M)$ on paper, it's not perfect by making a finite-sample approximation using limited number of task instances. For our experiments, $n_M\le 40$, which may induce a significant approximation error according to Eqn 11. This explains why in most of our experiments, UNICORN-SUP is inferior to UNICORN-SS. We believe that with more tasks and data, its performance can be improved.

Q5. Given the mathematical resemblence/equivalence of COMRL methods, why does UNICORN achieves suprior performance?

Compared to FOCAL and CORRO, directly optimizing $I(Z; M)$ or a convex combination of the primary and lesser causalities enforce a better trade-off between spurious correlation and contributory causality, see our response to Q2.

The comparison between UNICORN and CSRO is trikier, since the objective of CSRO and UNICORN-SS are mathematically equivalent. We believe the main difference lies in the implementation choices. CSRO approximates $I(Z; X_b)$ with the CLUB loss. On top of this approximation, it employs variational distribution $q_{\theta}(z|x_b)$ to approximate the true posterior $q(z|x_b)$ in Eqn 6. Therefore it involves $\textit{2 levels}$ of approximation. In contrast, both UNICORN variants involves only ${1 level}$ of approximation (finite sample approximation in Thm 2.4 for UNICORN-SUP and variational approximation in eqn 14 for UNICORN-SS). We believe this explains (at least partially) the superiority of UNICORN over CSRO.

Q6. Additional information (e.g., proof sketch, interpretation of Thm 2.4, experiments on discrete spaces, more related work...)

We appreciate all the suggestions and will provide these information using the 1 extra page should the paper be accepted.

---

### Author Response · Authors · 2024-08-07
**References**

[1] http://media.acc.qcc.cuny.edu/faculty/volchok/causalMR/CausalMR3.html

[2] MetaDiffuser: Diffusion Model as Conditional Planner for Offline Meta-RL

[3] ContraBAR: Contrastive Bayes-Adaptive Deep RL

[4] Varibad: A very good method for bayes-adaptive deep rl via meta-learning

---

### Decision · Program_Chairs · 2024-09-25

**Decision:**

Accept (spotlight)

**Comment:**

This paper provides a theoretical analysis that explains various offline meta-RL approaches in a unified way and proposes a new algorithm that outperforms the baselines across a broad range of offline meta-RL benchmarks. All of the reviewers unanimously found the theoretical analysis novel and insightful, as it unifies seemingly different approaches. They also found that the empirical results are strong and convincing enough. I believe that the findings of this paper would be interesting to the meta-RL community. Thus, I recommend to accept this paper.